# Revisiting Node Affinity Prediction in Temporal Graphs

**Or Feldman**[1][*]   **Krishna Sri Ipsit Mantri**[2][†][*]   **Moshe Eliasof**[1,3][‡]   **Chaim Baskin**[1][‡]

[1]Ben-Gurion University of the Negev, Israel
[2]Purdue University, USA
[3]University of Cambridge, UK

## Abstract

Node affinity prediction is a common task that is widely used in temporal graph learning with applications in social and financial networks, recommender systems, and more. Recent works have addressed this task by adapting state-of-the-art dynamic link property prediction models to node affinity prediction. However, simple heuristics, such as Persistent Forecast or Moving Average, outperform these models. In this work, we analyze the challenges in training current Temporal Graph Neural Networks for node affinity prediction and suggest appropriate solutions. Combining the solutions, we develop NAVIS - Node Affinity prediction model using Virtual State, by exploiting the equivalence between heuristics and state space models. While promising, training NAVIS is non-trivial. Therefore, we further introduce a dedicated loss function for node affinity prediction. We evaluate NAVIS on TGB and show that it outperforms the state-of-the-art, including heuristics. Our source code is available at `https://github.com/orfeld415/NAVIS`

## 1 Introduction

Temporal graphs provide a natural way to represent evolving interactions in systems such as trade networks, recommender systems, social platforms, and financial transactions (Kumar et al., 2019; Shetty & Adibi, 2004; Huang et al., 2023). A central challenge in this setting is *future node affinity prediction*: forecasting how strongly a node will interact with other nodes at a future time. This *differs* from future link prediction, which instead asks whether a particular edge will appear. In contrast, *affinity prediction* requires producing a full ranking over potential neighbors, making it more demanding but also more relevant to many real-world applications (MacDonald et al., 2015; Bertin-Mahieux et al., 2011; Nadiri & Takes, 2022; Shamsi et al., 2022).

In the context of link-level prediction, recent progress in Temporal Graph Neural Networks (TGNNs), including TGN (Rossi et al., 2020), TGAT (Xu et al., 2020), DyGFormer (Yu et al., 2023), and GraphMixer (Cong et al., 2023), has improved state-of-the-art performance. These methods rely on local neighborhood sampling and nonlinear message-passing, which are effective for future link prediction task. *However*, when applied to node *affinity prediction*, it is evident that they perform worse than simple heuristics such as Persistent Forecast and Moving Average (Huang et al., 2023). This gap suggests that current TGNNs designs do not align well with the inductive biases required for affinity prediction.

Accordingly, this paper aims to answer the following questions: *why do heuristics outperform more sophisticated TGNNs for future node affinity prediction? and can we push TGNNs to do better?* We argue that the advantage arises from a confluence of factors and identify several contributing issues that collectively explain this phenomenon, including: (i) **Expressivity.** Existing TGNNs cannot represent a simple Moving Average of past affinities, because their nonlinear updates and reliance on sampled neighborhoods prevent them from maintaining the required linear memory. (ii) **Loss mismatch.** Cross-entropy, commonly used as a loss function for link prediction, is not well-aligned with the

---

[*]Equal contribution.
[†]Work done while interning at the INSIGHT Lab, Ben-Gurion University of the Negev.
[‡]Equal supervision.

ranking nature of affinity tasks. (iii) **Global temporal dynamics.** Affinities often depend on shared network-wide trends (e.g., regime shifts), which local sampling does not capture. (iv) **Information loss.** TGNNs are broadly categorized into memory-based and non-memory-based architectures. Memory-based models (e.g., TGN (Rossi et al., 2020) and DyRep (Trivedi et al., 2019)) maintain per-node states; however, batch processing of events can cause short-term updates within a batch to be missed (Feldman & Baskin, 2024). In contrast, non-memory-based methods recompute embeddings from scratch at prediction time and maintain no states, making them prone to overlooking earlier updates in the evolving graph (Cong et al., 2023; Yu et al., 2023).

These observations guide and motivate our work. Our key idea is that heuristics like persistent Forecast and Moving Average are not arbitrary fixes, but rather special cases of *linear state space models* (SSMs) (Gu et al., 2021; 2022; Ceni et al., 2025; Eliasof et al., 2025), which naturally provide memory and long-range temporal dependencies. By embedding the structure of SSMs into a learnable TGNN, we can retain the robustness of heuristics while extending their expressivity.

Building on this idea, we introduce NAVIS (Node Affinity prediction with Virtual State). NAVIS maintains both per-node state and a virtual global state that co-evolve with the dynamic graph structure, thereby providing a principled memory mechanism suitable for the requirements of future node affinity prediction. Additionally, to address the loss mismatch, we propose a rank-based objective that is better suited to ordinal affinity outputs. Importantly, we do not claim to have solved the problem entirely: the approach still inherits limitations, for example, in modeling complex multi-hop dependencies. Nonetheless, our results indicate meaningful progress that can shape the future of node affinity prediction.

To illustrate the existing challenges and the effectiveness of NAVIS, we include a synthetic node-affinity experiment in Figure 1. In this controlled setting, node affinities depend on a hidden global process with nonlinear structure. Although simple heuristics outperform state-of-the-art TGNNs, since they capture only per-node history, as TGNNs, they fail to recover the shared latent space. By maintaining a virtual global state, NAVIS achieves the lowest error. Although simplified, this example highlights the importance of global temporal dynamics and motivates one of the design choices in NAVIS. We provide the full experiment details in Appendix C.

**Our contributions.**

1. We theoretically show that simple heuristics are special cases of linear SSMs, and use this connection to design a TGNN architecture that generalizes them, making it more expressive.

2. We analyze why cross-entropy loss is suboptimal for affinity prediction, and develop a rank-based alternative that improves optimization and aligns with evaluation metrics.

3. We provide extensive experiments on the Temporal Graph Benchmark (TGB) and additional datasets, demonstrating consistent improvements over both heuristics and prior TGNNs. The significance of our experiments is that they validate the importance of aligning model inductive biases and training objectives with the task.

## 2 BACKGROUND

In this section, we provide essential information related to our work, from basic notations and definitions, to simple future node affinity prediction baselines, which we generalize in NAVIS.

**Notations and Definitions.** We consider a *continuous-time dynamic graph* (CTDG) as a stream of timestamped interactions between ordered node pairs drawn from the node set $\mathcal{V} = \{1, \ldots, n\}$. The CTDG observed up to time $t$ is:

$$\mathcal{G}_t = \{(u_j, v_j, \tau_j, w_j)\}_{j=1}^{J(t)}, \tag{1}$$

where $u_j, v_j \in \mathcal{V}$ denote source and target nodes, $\tau_j \in \mathbb{R}^+$ is the interaction time, $w_j \in \mathbb{R}$ is its weight and $J(t)$ is the number of interactions occurred up to time $t$. The *future node affinity prediction* problem seeks, for a query node $u \in \mathcal{V}$ and a future time $t^+ > t$, to estimate the node's affinity to every other node $v \in \mathcal{V} \setminus \{u\}$ conditioned on $\mathcal{G}_t$. A parameterized model $F_{\boldsymbol{\theta}}$ produces the predicted affinity scores vector

$$\mathbf{s} = F_{\boldsymbol{\theta}}(u, \mathcal{G}_t, t^+) \in \mathbb{R}^{|\mathcal{V}|}. \tag{2}$$

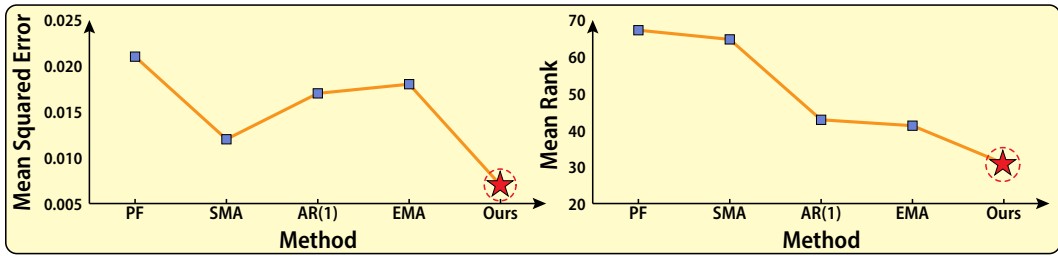

Figure 1: **Synthetic node affinity experiment.** Node affinities depend on a global, regime-switching latent $g(t)$ with nonlinear component $g(t)^2$ and node-specific phases. Baselines relying only on per-node histories (PERSISTENT FORECAST, SMA, EMA) or a local AR(1) SSM cannot recover the shared latent space, leading to higher error. OURS, indicating NAVIS, maintains a virtual global state, achieves the lowest error on both metrics. In Appendix C we provide the full experiment details and baseline descriptions.

Given the ground truth affinities $\mathbf{y}$ realized at $t^+$, we learn $\boldsymbol{\theta}$ by minimizing a task-specific loss $\ell$:

$$\min_{\boldsymbol{\theta}} \sum_{u \in \mathcal{V}} \ell\left(F_{\boldsymbol{\theta}}(u, \mathcal{G}_t, t^+), \mathbf{y}\right). \tag{3}$$

**Historical Average.** A simple interaction-level baseline is the historical average, which estimates each source–destination affinity by the mean weight of all past interactions observed prior to $t^+$:

$$\mathbf{s}(v) = \frac{1}{\#u, v} \sum_{(u,v,\tau_j,w_j) \in \mathcal{G}_t} w_j. \tag{4}$$

Here $\#_{u,v}$ denotes the number of observed $(u, v)$ interactions up to time $t$.

**Moving Average and State Space Models.** Allowing the tested models to use previous ground truth affinity vectors at inference, instead of the full fine-grained CTDG, creates additional schemes to predict the future affinity vectors. The *Persistent Forecast* (PF) heuristic is the most basic one that utilizes previous affinity vectors. PF outputs the previous affinity vector as future prediction, i.e.:

$$\mathbf{s} = \mathbf{x} \tag{5}$$

where $\mathbf{x}$ is the previous (most recent) ground truth affinity vector.

Another natural vector-level heuristic for future node affinity prediction is the *Exponential Moving Average* (EMA), which maintains an estimate of a node's affinity vector by exponentially weighting recent affinity vectors:

$$\mathbf{s} = \mathbf{h}_i = \alpha \mathbf{h}_{i-1} + (1 - \alpha)\mathbf{x} \tag{6}$$

here, $\alpha \in [0, 1]$ is the decay parameter and $\mathbf{h}_i, \mathbf{h}_{i-1}$ are hidden states. Note that PF is a specific case of EMA where $\alpha = 0$.

An alternative is the *Simple Moving Average* (SMA) with window size $w$, which averages over a finite window. Its recursive form is:

$$\mathbf{s} = \mathbf{h}_i = \frac{w-1}{w}\mathbf{h}_{i-1} + \frac{1}{w}\mathbf{x}. \tag{7}$$

EMA and SMA use an infinite geometric decay, retaining long but diminishing memory. Although these filters can capture long-term dynamics, they are limited to fixed, hand-crafted memory kernels. To move beyond such ad-hoc designs, we can view EMA through the lens of latent dynamical systems.

Departing from simple averaging approaches, State Space Models (SSMs) provide a principled framework to model temporal sequences via hidden state evolution and observation processes (Hamilton, 1994; Aoki, 2013). A discrete linear SSM is defined as:

$$\mathbf{h}_i = \mathbf{A}\mathbf{h}_{i-1} + \mathbf{B}\mathbf{x}, \tag{8}$$

$$\mathbf{s} = \mathbf{C}\mathbf{h}_i + \mathbf{D}\mathbf{x}, \tag{9}$$

Where $\mathbf{A}, \mathbf{B}, \mathbf{C}, \mathbf{D}$ are learnable matrices.

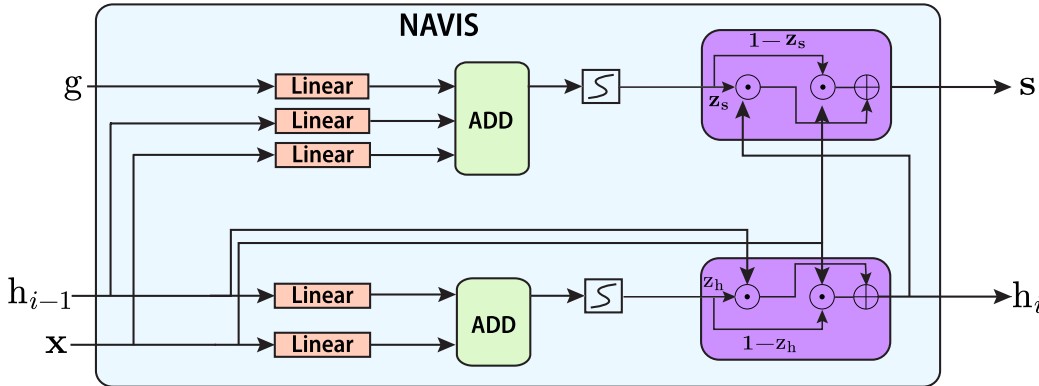

Figure 2: NAVIS architecture for node affinity prediction. The current state and previous affinity vector are projected through linear transformations and aggregated into a new state. A lightweight gated mechanism ensures a persistent, linear input–output. The predicted affinity vector is then produced directly from this state based on the virtual global state.

**Beyond Moving Average.** We show that vector-level heuristics are instances of SSMs. This reveals a clear hierarchy in model expressiveness, where SSMs are more expressive, generalizing Moving Average.

**Theorem 1** (Linear SSMs generalize basic heuristics). *Let $\mathcal{H}$ be the set of basic heuristics (PF, SMA, EMA), $\mathcal{F}_{\text{lin-SSM}}$ be the set of maps realizable by the linear SSM in Equation 8 and Equation 9. Then, the following strict inclusion holds:*

$$\mathcal{H} \subsetneq \mathcal{F}_{\text{lin-SSM}}. \tag{10}$$

*Proof.* $\mathcal{H} \subsetneq \mathcal{F}_{\text{lin-SSM}}$. First, we show containment ($\subseteq$). Each heuristic corresponds to a specific choice of $(\mathbf{A}, \mathbf{B}, \mathbf{C}, \mathbf{D})$:

- EMA($\alpha$): $\mathbf{A} = \alpha \mathbf{I}$, $\mathbf{B} = (1 - \alpha)\mathbf{I}$, $\mathbf{C} = \mathbf{I}$, $\mathbf{D} = 0$.

- SMA($w$): $\mathbf{A} = \frac{w-1}{w}\mathbf{I}$, $\mathbf{B} = \frac{1}{w}\mathbf{I}$, $\mathbf{C} = \mathbf{I}$, $\mathbf{D} = 0$.

- PF: $\mathbf{A} = 0$, $\mathbf{B} = 0$, $\mathbf{C} = 0$, $\mathbf{D} = I$.

Next, we show the inclusion is strict ($\subsetneq$). Consider a set of $2 \times 2$ SSM matrices with $\mathbf{A} = \text{diag}(\alpha_1, \alpha_2)$, $\mathbf{B} = \text{diag}(1 - \alpha_1, 1 - \alpha_2)$, $\mathbf{C} = \mathbf{I}$, $\mathbf{D} = 0$ where $\alpha_1 \neq \alpha_2$. These models' weights on past inputs cause each entry in the affinity vector to decay at a different rate. This behavior cannot be replicated by the single decay of an EMA. Thus, $\mathcal{F}_{\text{lin-SSM}} \setminus \mathcal{H} \neq \varnothing$. $\square$

**Implications.** This hierarchy that stems from Theorem 1 reveals the potential of SSMs: although simple heuristics outperform any state-of-the-art TGNN, a carefully designed SSM-based TGNN can potentially outperform these heuristics because it generalizes them.

## 3 METHOD

This section contains our key contributions and is organized as follows: in Section 3.1 we detail factors that hinder TGNNs performance for future node affinity prediction. Motivated by these findings, in Section 3.2 we introduce our model. In Section 3.3 we detail how to train it effectively.

### 3.1 WHAT HINDERS TGNNS IN FUTURE NODE AFFINITY PREDICTION

As established in Section 2, common heuristics like PF and EMA are special cases of SSMs. This advantage of SSMs over heuristics is achieved due to the fact that SSMs can output a linear combination of the previous affinity vector and the previous predicted affinity vector. We now discuss

another important theoretical direction: RNN (Elman, 1990), LSTM (Hochreiter & Schmidhuber, 1997) or GRU (Cho et al., 2014) cells — that are commonly used in popular memory-based TGNNs (Trivedi et al., 2019; Rossi et al., 2020; Tjandra et al., 2024) — cannot express the most basic heuristic of PF, thereby hindering memory-based TGNNs performance.

**Theorem 2.** *Let $\{h_i\}_{i\geq0}$ be the hidden states generated by a single standard RNN cell, LSTM cell, or GRU cell driven by inputs $\{x_i\}_{i\geq1} \subseteq \mathbb{R}^d$. There do not exist parameters of these cells such that, for all $t$ and all input sequences, $h_i = x_i$ (PF).*

*Proof.* We assume the standard elementwise nonlinearities $\sigma(u) = \frac{1}{1+e^{-u}} \in (0, 1)$ and $\tanh(u) \in (-1, 1)$. Below, we present the equations that define common recurrent models.

$$\text{(RNN)} \quad \mathbf{h}_i = \phi(W_h\mathbf{h}_{i-1} + W_x x_i + \mathbf{b}), \quad \phi = \tanh. \tag{11}$$

$$\text{(LSTM)} \quad \begin{aligned} \mathrm{i}_i &= \sigma(W_i[\mathbf{h}_{i-1}; \mathbf{x}_i] + \mathbf{b}_i), & \mathbf{f}_i &= \sigma(W_f[\mathbf{h}_{i-1}; \mathbf{x}_i] + \mathbf{b}_f), \\ \mathrm{o}_i &= \sigma(W_o[\mathbf{h}_{i-1}; \mathbf{x}_i] + \mathbf{b}_o), & \mathbf{g}_i &= \tanh(W_g[\mathbf{h}_{i-1}; \mathbf{x}_i] + \mathbf{b}_g), \\ \mathbf{c}_i &= \mathbf{f}_i \odot \mathbf{c}_{i-1} + \mathrm{i}_i \odot \mathbf{g}_i, & \mathbf{h}_i &= \mathrm{o}_i \odot \tanh(\mathbf{c}_i). \end{aligned} \tag{12}$$

$$\text{(GRU)} \quad \begin{aligned} \mathbf{z}_i &= \sigma(W_z[\mathbf{h}_{i-1}; \mathbf{x}_i] + \mathbf{b}_z), & \mathbf{r}_i &= \sigma(W_r[\mathbf{h}_{i-1}; \mathbf{x}_i] + \mathbf{b}_r), \\ \tilde{\mathbf{h}}_i &= \tanh\big(W[\mathbf{r}_i \odot \mathbf{h}_{i-1}; \mathbf{x}_i] + \mathbf{b}\big), \\ \mathbf{h}_i &= (1 - \mathbf{z}_i) \odot \mathbf{h}_{i-1} + \mathbf{z}_i \odot \tilde{\mathbf{h}}_i. \end{aligned} \tag{13}$$

We show that no choice of parameters in RNN, LSTM or GRU yields the map $\mathbf{h}_i = \mathbf{x}_i$.

*RNN:* $\mathbf{h}_t = \tanh(W_h\mathbf{h}_{i-1} + W_x\mathbf{x}_i + \mathbf{b})$ takes values in $(-1, 1)^d$, while the mapping $\mathbf{x}_t \mapsto \mathbf{h}_t$ is unbounded on $\mathbb{R}^d$. Hence, equality $\forall\mathbf{x}_i$ is impossible.

*LSTM:* $\mathbf{h}_i = \mathrm{o}_i \odot \tanh(\mathbf{c}_i)$ with $\mathrm{o}_i \in (0, 1)^d$ and $\tanh(\mathbf{c}_i) \in (-1, 1)^d$ implies $\mathbf{h}_i \in (-1, 1)^d$, again contradicting the unbounded range of $\mathbf{h}_i$.

*GRU:* Set $\mathbf{h}_{i-1} = 0$. Then $\mathbf{h}_i = \mathbf{z}_i \odot \tilde{\mathbf{h}}_i$ with $\mathbf{z}_i \in (0, 1)^d$ and $\tilde{\mathbf{h}}_i = \tanh(\cdot) \in (-1, 1)^d$, hence $\mathbf{h}_i \in (-1, 1)^d$ cannot equal $\mathbf{x}_i$ for arbitrary $\mathbf{x}_i \in \mathbb{R}^d$. $\qquad\square$

Theorem 2 has profound implications: any memory-based TGNN that applies standard memory cells cannot represent even the simplest heuristic – Persistent Forecasting (PF), which have been proven empirically to perform exceptionally well on node affinity tasks. This theoretical result is at the underpinnings of our work, and motivates us to generalize heuristics while remaining more expressive. Thus, we design NAVIS as a simple learnable linear SSM. Tjandra et al. (2024) showed that identifying the target node when updating the node state of the source node is a necessary property. Without this property, TGNNs cannot express the persistent forecast heuristic. In the proof of Theorem 2, we explicitly assume that the target node is identified via its corresponding index in the affinity vector of the source node. Hence, Theorem 2 holds even when the target node is identified, revealing another necessary condition for expressing the persistent forecast heuristic.

**Current TGNNs Underutilize Available Temporal Information**

We argue that a major source of empirical underperformance in TGNNs is the systematic loss of temporal information. Memory–based TGNNs often rely on batching for tractable runtimes; however, batching can obscure multiple interactions that affect the same node within a single batch window, thereby dropping intermediate state transitions (Feldman & Baskin, 2024). In contrast, non–memory architectures, e.g., DyGFormer (Yu et al., 2023), GraphMixer (Cong et al., 2023), and DyGMamba (Ding et al., 2025), avoid within-batch omissions by maintaining a buffer of recent events and recomputing node embeddings on demand. To bound latency, these buffers are fixed in size; once filled, the oldest events are evicted, discarding potentially informative long-term interaction events. This hard truncation differs from EMA, where the influence of older events decays but remains non-zero, allowing previous interaction events to shape future affinity predictions. A further, underexploited source of information is the evolving global graph state. Memory–based TGNNs

typically use few message-passing layers per state update, limiting the incorporation of broader context (Rossi et al., 2020), while non–memory methods commonly restrict buffered events to 1-hop neighborhoods. Notably, also recent sophisticated state-of-the-art TGNNs do not utilize the full global state (Lu et al., 2024; Gravina et al., 2024). We leverage these observations to design a TGNN that preserves fine-grained temporal transitions, retains long-term interaction events, and integrates global graph context to maximize the use of available information to give an accurate prediction for future node affinity.

## 3.2 NAViS: Node Affinity Prediction with a Global Virtual State

Motivated by Section 3.1, we propose NAViS— a node-affinity prediction model that utilizes a linear state-space mechanism to maintain a state $\mathbf{h} \in \mathbb{R}^d$ for each node, and a virtual global state $\mathbf{g} \in \mathbb{R}^d$, where $d = |\mathbb{V}|$ denotes the affinity-space dimension. Transitions are computed by a learnable linear SSM that enforces the output to be a linear combination of the inputs, akin to an EMA, but with flexibility to allow the coefficient $\alpha$ to be computed at runtime from the current events rather than being fixed. Concretely, we define NAViS as the following sequence of update steps:

$$
\begin{aligned}
\mathbf{z}_h &= \sigma(W_{xh}\mathbf{x} + W_{hh}\mathbf{h}_{i-1} + \mathbf{b}_h), \\
\mathbf{h}_i &= \mathbf{z}_h \odot \mathbf{h}_{i-1} + (1 - \mathbf{z}_h) \odot \mathbf{x}, \\
\mathbf{z}_s &= \sigma(W_{xs}\mathbf{x} + W_{hs}\mathbf{h}_i + W_{gs}\mathbf{g} + \mathbf{b}_s), \\
\mathbf{s} &= \mathbf{z}_s \odot \mathbf{h}_i + (1 - \mathbf{z}_s) \odot \mathbf{x},
\end{aligned}
\tag{14}
$$

where $\mathbf{x}, \mathbf{h}_{i-1}, \mathbf{h}_i, \mathbf{g}, \mathbf{s} \in \mathbb{R}^d$ are the previous affinity vector, previous node state, updated node state, virtual global vector, and predicted affinity vector, respectively. The parameters $W_{xh}, W_{hh}, W_{xs}, W_{hs}, W_{gs} \in \mathbb{R}^{1 \times d}$ are learnable weights, $\mathbf{b}_h, \mathbf{b}_s \in \mathbb{R}^d$ are learnable biases, and $\sigma$ is the sigmoid function, forcing $\mathbf{z}_h, \mathbf{z}_s$ to be in $[0, 1]$, to maintain conceptual similarity with $\alpha$ in Equation (6). In Figure 2 we provide a detailed scheme of NAViS.

We compute the virtual global vector $\mathbf{g}$ by maintaining a buffer of the most recent previous affinity vectors, globally. Then, the virtual global vector is computed by performing aggregation over all the vectors in the buffer. The goal of the global vector is to detect a global trend (e.g, a new song or a new TV series that is globally streamed) before we are queried about a specific node. In practice, aggregating the buffer with the most recent vector selection is efficient and empirically effective, as we show later in Section 4.

**Handling a full CTDG.** When a previous affinity vector is unavailable and predictions must rely solely on interaction weights of the CTDG, we estimate the previous affinity vector of $u$, $\mathbf{x}$, via $\hat{\mathbf{x}}$. We initialize $\hat{\mathbf{x}} = \mathbf{0}$ and, upon each weighted interaction between source node $u$ and destination node $v$, add the interaction weight to the $v$-th entry of $\hat{\mathbf{x}}$. When $\hat{\mathbf{x}}$ is required for prediction, we normalize it by dividing by the sum of its entries. After computing the future affinity vector of $u$ we again set $\hat{\mathbf{x}} = \mathbf{0}$.

**Key Properties of NAViS.** We note that NAViS generalizes EMA and other heuristics by allowing the gates $\mathbf{z}_h$ and $\mathbf{z}_s$ to adapt to new information, in contrast to a fixed $\alpha$. Large gate values enable the model to retain long-term information when beneficial. Notably, NAViS does not rely on neighbors' hidden states, unlike other memory-based TGNNs, and therefore is compatible with the t-Batch mechanism (Kumar et al., 2019), enabling efficient batching without missing updates. In addition, we show in Appendix F that incorporating global information via $\mathbf{g}$ can improve the accuracy of the predicted affinity vector when global trends affect the nodes' affinities.

**NAViS for large-scale graphs.** For graphs with $N$ nodes, the number of learnable parameters in NAViS scales as $O(N)$, which can be prohibitive for graphs with millions of nodes. To make NAViS practical at this scale, we introduce a sparsified affinity prediction pipeline. Specifically, for each node we retain only the entries corresponding to candidate target nodes. In real-world settings, such as streaming services where users and movies are nodes and interactions are edges, we are interested in each user's affinity to movies rather than to other users. In practice, this substantially reduces the parameter count of NAViS. For example, on the tgbn-token dataset (Shamsi et al., 2022), which records user–token interactions, NAViS requires about (5,000) parameters, while the graph contains over (60,000) nodes. We further provide a detailed empirical runtime and memory analysis of NAViS in Appendix G.

### 3.3 Learning with Rank-Based Loss: Why Cross-Entropy Fails

With NAVIS specified in Section 3.2, we now turn to the question of *how to train it effectively*. Because most downstream uses of affinity vectors depend on the induced ordering of candidates rather than the actual affinity values (Huang et al., 2023; Tjandra et al., 2024), the choice of the loss function is critical. Most TGNNs use the cross-entropy loss (Luo & Li, 2022; Yu et al., 2023; Tjandra et al., 2024), which treats the output as a categorical distribution and ignores ordinal structure. As we show next, this property is suboptimal, and requires a designated loss to address this limitation.

**The Limitation of Cross-Entropy Loss.** Let $\mathbf{y}, \mathbf{s} \in \mathbb{R}^d$ be the ground-truth and predicted affinity vectors. The cross-entropy loss reads:

$$\ell_{\text{CE}}(\mathbf{s}, \mathbf{y}) = -\sum_{v \in \mathbb{V}} \mathbf{y}(v) \log[\text{softmax}(\mathbf{s})](v). \tag{15}$$

By construction, this loss is suboptimal because it penalizes well-ranked predictions with mismatched magnitudes. We formalize this claim in Theorem 3.

**Theorem 3** (Cross-Entropy is Suboptimal for Ranking). *There exist infinitely many triplets of $\mathbf{y}$, a ground-truth affinity vector, and $\mathbf{s}_1$, $\mathbf{s}_2$, two predicted logits of affinity vectors such that: $\mathbf{s}_1$ ordering exactly matches the ground-truth node affinity scores (i.e., yields a perfect ordering), $\mathbf{s}_2$ differs from $\mathbf{s}_1$ only in the logit of the most-affinitive node or the least-affinitive node—thereby making the ordering of $\mathbf{s}_2$ different from $\mathbf{s}_1$ and $\mathbf{y}$, and $\ell_{\text{CE}}(\mathbf{s}_1, \mathbf{y}) > \ell_{\text{CE}}(\mathbf{s}_2, \mathbf{y})$.*

*Proof.* Set $\mathbf{y} = [0.4, 0.6]$, $\mathbf{s}_1 = [1, 2.4]$ (correctly rank logits), $\mathbf{s}_2 = [1, 0.6]$ (wrongly ranked logits). Then $0.75 = \ell_{\text{CE}}(\mathbf{s}_2, \mathbf{y}) < \ell_{\text{CE}}(\mathbf{s}_1, \mathbf{y}) = 0.78$, where $\mathbf{s}_1$ ranks the same as $\mathbf{y}$ while $\mathbf{s}_2$ does not. Since $\ell_{\text{CE}}$ is a continuous function, there are infinitely many such triplets. $\square$

To address this shortcoming, we train NAVIS using *Lambda Loss* (Burges et al., 2006) that is defined as follows:

$$\ell_{\text{Lambda}}(\mathbf{s}, \mathbf{y}) = \sum_{y_i > y_j} \log_2\left(\frac{1}{1 + e^{-\sigma\left(\mathbf{s}_{\pi_i} - \mathbf{s}_{\pi_j}\right)}}\right) \delta_{ij} \left|A_{\pi_i} - A_{\pi_j}\right|, \tag{16}$$

where $\pi_i$ is the index of the node at rank $i$ after sorting the affinity scores and $A_{\pi_i}, \delta_{ij}, D_i$ are defined as:

$$A_{\pi_i} = \frac{2^{y_{\pi_i}} - 1}{\text{maxDCG}}, \qquad \delta_{ij} = \left|\frac{1}{D_{|i-j|}} - \frac{1}{D_{|i-j|+1}}\right|, \qquad D_i = \log_2(1 + i) \tag{17}$$

and $\text{maxDCG}$ is the maximum Discounted Cumulative Gain (DCG) computed by: $\max_{\pi'} \sum_{i=1}^{d} \frac{y_{\pi'(i)}}{\log_2(i+1)}$. The loss in Equation (16) was previously shown to be effective for ranking tasks (Burges et al., 2011; Wang et al., 2018). This loss directly optimizes rank-based objectives via pairwise "lambdas" that approximate the gradient of non-differentiable ranking-based metrics, focusing learning on swaps that most impact the final ranking.

**Pairwise Margin Regularization.** In our experiments, we discovered that the use of the loss in Equation (16) alone is not sufficient for future node affinity prediction tasks, as we elaborate in Appendix F. Hence, we suggest the following regularization:

$$\ell_{\text{Reg}}(\mathbf{s}, \mathbf{y}) = \sum_{y_i > y_j} \max(0, -(\mathbf{s}_{\pi_i} - \mathbf{s}_{\pi_j}) + \Delta), \tag{18}$$

Here, $\Delta$ is a hyperparameter that represents the minimum margin required between each pair of affinity scores. The goal of this regularization is to prevent NAVIS from incorrectly learning to shrink the affinity scores to minimize Equation (16). In Appendix H we show that the undesirable property in Theorem 3 does not hold for the suggested loss function.

## 4 Experiments

We evaluate NAVIS across multiple future node affinity prediction benchmarks and compare it with recent state-of-the-art baselines, including both heuristics and TGNNs. Sections 4.1 and 4.2 present

Table 1: NDCG@10 on TGB datasets (↑ higher is better). NAVIS is benchmarked against TGNNs that use all available graph messages. Boldface marks the best method.

| Method | tgbn-trade | | tgbn-genre | | tgbn-reddit | | tgbn-token | |
|---|---|---|---|---|---|---|---|---|
| | Val. | Test | Val. | Test | Val. | Test | Val. | Test |
| Moving Avg | 0.793 | 0.777 | 0.496 | 0.497 | 0.498 | 0.480 | 0.401 | 0.414 |
| Historical Avg | 0.793 | 0.777 | 0.478 | 0.472 | 0.499 | 0.481 | 0.402 | 0.415 |
| JODIE | $0.394\pm0.05$ | $0.374\pm0.09$ | $0.358\pm0.03$ | $0.350\pm0.04$ | $0.345\pm0.02$ | $0.314\pm0.01$ | – | – |
| TGAT | $0.395\pm0.14$ | $0.375\pm0.07$ | $0.360\pm0.04$ | $0.352\pm0.03$ | $0.345\pm0.01$ | $0.314\pm0.01$ | – | – |
| CAWN | $0.393\pm0.07$ | $0.374\pm0.09$ | – | – | – | – | – | – |
| TCL | $0.394\pm0.11$ | $0.375\pm0.09$ | $0.362\pm0.04$ | $0.354\pm0.02$ | $0.347\pm0.01$ | $0.314\pm0.01$ | – | – |
| GraphMixer | $0.394\pm0.17$ | $0.375\pm0.11$ | $0.361\pm0.04$ | $0.352\pm0.03$ | $0.347\pm0.01$ | $0.314\pm0.01$ | – | – |
| DyGFormer | $0.408\pm0.58$ | $0.388\pm0.64$ | $0.371\pm0.06$ | $0.365\pm0.20$ | $0.348\pm0.02$ | $0.316\pm0.01$ | – | – |
| DyGMamba | $0.393\pm0.001$ | $0.374\pm0.001$ | $0.359\pm0.001$ | $0.351\pm0.001$ | $0.347\pm0.000$ | $0.314\pm0.000$ | – | – |
| DyRep | $0.394\pm0.001$ | $0.374\pm0.001$ | $0.357\pm0.001$ | $0.351\pm0.001$ | $0.344\pm0.001$ | $0.312\pm0.001$ | $0.151\pm0.006$ | $0.141\pm0.006$ |
| TGN | $0.445\pm0.009$ | $0.409\pm0.005$ | $0.443\pm0.002$ | $0.423\pm0.007$ | $0.482\pm0.007$ | $0.408\pm0.006$ | $0.251\pm0.000$ | $0.200\pm0.005$ |
| TGNv2 | $0.807\pm0.006$ | $0.735\pm0.006$ | $0.481\pm0.001$ | $0.469\pm0.002$ | $0.544\pm0.000$ | $0.507\pm0.002$ | $0.321\pm0.001$ | $0.294\pm0.001$ |
| NAVIS (ours) | $\mathbf{0.872}\pm0.001$ | $\mathbf{0.863}\pm0.001$ | $\mathbf{0.512}\pm0.001$ | $\mathbf{0.520}\pm0.001$ | $\mathbf{0.564}\pm0.001$ | $\mathbf{0.552}\pm0.001$ | $\mathbf{0.423}\pm0.001$ | $\mathbf{0.444}\pm0.001$ |

Table 2: NDCG@10 on TGB datasets using only previous ground-truth labels (↑ higher is better). This setting is suited for heuristics. Boldface marks the best method. Baselines have no standard deviation because they are pre-defined and deterministic.

| Method | tgbn-trade | | tgbn-genre | | tgbn-reddit | | tgbn-token | |
|---|---|---|---|---|---|---|---|---|
| | Val. | Test | Val. | Test | Val. | Test | Val. | Test |
| Persistent Forecast | 0.860 | 0.855 | 0.350 | 0.357 | 0.380 | 0.369 | 0.403 | 0.430 |
| Moving Avg | 0.841 | 0.823 | 0.499 | 0.509 | 0.574 | 0.559 | 0.491 | 0.508 |
| NAVIS (ours) | $\mathbf{0.872}\pm0.001$ | $\mathbf{0.863}\pm0.001$ | $\mathbf{0.517}\pm0.001$ | $\mathbf{0.528}\pm0.001$ | $\mathbf{0.584}\pm0.001$ | $\mathbf{0.569}\pm0.001$ | $\mathbf{0.493}\pm0.001$ | $\mathbf{0.513}\pm0.001$ |

our key empirical findings, and Appendix F provides additional ablation studies. Our experiments aim to address the following research questions: **(RQ1)** How does NAVIS perform compared to prior art for future node affinity prediction? **(RQ2)** Does our method generalize across various types of graphs? **(RQ3)** What is the contribution of each component in NAVIS?

**Experimental setup** We compare NAVIS to the following TGNN baselines JODIE(Kumar et al., 2019), TGAT(Xu et al., 2020), CAWN (Wang et al., 2021b), TCL (Wang et al., 2021a), Graph-Mixer(Cong et al., 2023),DyGFormer(Yu et al., 2023),DyRep(Trivedi et al., 2019), TGN(Rossi et al., 2020), TGNv2(Tjandra et al., 2024) and the standard heuristics presented in (Huang et al., 2023) and in (Tjandra et al., 2024). Following the standard protocols (Huang et al., 2023), we use a 70%-15%-15% chronological split, train for 50 epochs, use a batch size of 200, and report the average NDCG@10 (Järvelin & Kekäläinen, 2002) over three runs. We include both future node affinity prediction settings: (1) using the full fine-grained CTDG up to the prediction time, and (2) using only previous ground-truth affinity vectors. Previous TGNNs only support the first setting (Tjandra et al., 2024), and, therefore, only heuristics are included in the second setting comparisons.

## 4.1 NODE AFFINITY PREDICTION ON TGB

To answer **(RQ1)**, we use the TGB datasets for node affinity prediction (Huang et al., 2023) : `tgbn-trade`, `tgbn-genre`, `tgbn-reddit`, and `tgbn-token`. As shown in Tables 1 and 2, **NAVIS outperforms all baselines in both experimental settings**. It improves over the best-performing TGNN, TGNv2, by +12.8% on `tgbn-trade`. Notably, many TGNNs underperform simple heuristics, which aligns with our theoretical analysis that they are not optimized for ranking and underutilize available temporal information. In contrast, NAVIS linear design and rank-aware loss enable superior performance.

## 4.2 GENERALIZATION TO LINK PREDICTION DATASETS

To answer **(RQ2)**, we repurpose four temporal link prediction datasets (Wikipedia(Kumar et al., 2019), Flights(Strohmeier et al., 2021), USLegis(Fowler, 2006), and UNVote(Voeten et al., 2009)) for the future node affinity prediction task. We detail how we adjust these datasets in Appendix B. TGNNs are known to operate well on these datasets (Yu et al., 2023), and, therefore, should constitute a strong baseline. As shown in Tables 3 and 4, **NAVIS consistently outperforms both TGNN and

Table 3: NDCG@10 on converted link prediction datasets (↑ higher is better). NAVIS is benchmarked against TGNNs that use all available graph messages. Boldface marks the best method.

| Method | Wikipedia | | Flights | | USLegis | | UNVote | |
|---|---|---|---|---|---|---|---|---|
| | Val. | Test | Val. | Test | Val. | Test | Val. | Test |
| Historical Avg | 0.547 | 0.555 | 0.487 | 0.499 | 0.274 | 0.287 | 0.926 | 0.917 |
| Moving Avg | 0.547 | 0.555 | 0.029 | 0.028 | 0.150 | 0.154 | 0.926 | 0.918 |
| DyRep | 0.019±0.022 | 0.023±0.026 | 0.000±0.000 | 0.000±0.000 | 0.231±0.031 | 0.123±0.061 | 0.800±0.002 | 0.804±0.002 |
| DyGFormer | 0.058±0.002 | 0.058±0.002 | – | – | 0.271±0.036 | 0.220±0.057 | 0.817±0.007 | 0.809±0.005 |
| DyGMamba | 0.046±0.003 | 0.050±0.002 | – | – | 0.246±0.015 | 0.154±0.044 | 0.814±0.002 | 0.804±0.002 |
| TGN | 0.056±0.005 | 0.065±0.006 | 0.249±0.003 | 0.227±0.007 | 0.219±0.022 | 0.190±0.024 | 0.807±0.003 | 0.792±0.006 |
| TGNv2 | 0.478±0.005 | 0.433±0.004 | 0.326±0.008 | 0.299±0.014 | 0.323±0.036 | 0.253±0.040 | 0.824±0.008 | 0.813±0.010 |
| NAVIS (ours) | **0.564**±0.001 | **0.573**±0.001 | **0.489**±0.001 | **0.499**±0.001 | **0.331**±0.001 | **0.347**±0.001 | **0.969**±0.001 | **0.952**±0.001 |

Table 4: NDCG@10 on converted link prediction datasets using only previous ground-truth labels (↑ higher is better). Baselines have no standard deviation because they are pre-defined and deterministic. Boldface marks the best method.

| Method | Wikipedia | | Flights | | USLegis | | UNVote | |
|---|---|---|---|---|---|---|---|---|
| | Val. | Test | Val. | Test | Val. | Test | Val. | Test |
| Persistent Forecast | 0.499 | 0.507 | 0.296 | 0.307 | 0.328 | 0.320 | 0.963 | 0.917 |
| Moving Avg | 0.538 | 0.552 | 0.468 | 0.482 | 0.250 | 0.276 | 0.963 | 0.953 |
| NAVIS (ours) | **0.559**±0.001 | **0.566**±0.001 | **0.482**±0.001 | **0.494**±0.001 | **0.333**±0.001 | **0.326**±0.001 | **0.971**±0.001 | **0.953**±0.001 |

**heuristic baselines**, with gains ranging from +13.9% to +20% over the second performing TGNN, suggesting NAVIS generalizes well to many dynamic graph datasets. Similar to the results in Tables 1 and 2, other TGNNs fall behind simple heuristics even though they were shown to produce great results on these datasets for future link prediction tasks, strengthening that the TGNNs design choices mentioned earlier in Sections 3.1 and 3.3 are incompatible for future node affinity prediction.

### 4.3 ABLATION STUDY

To answer **(RQ3)**, we conduct an ablation study to isolate the contribution of each core component of NAVIS: (1) the linear state update mechanism (vs. the commonly used GRU), (2) the inclusion of the global virtual vector $\mathbf{g}$, and (3) the proposed loss (vs. cross-entropy, CE). Full results are provided in Appendix F. The ablations show that each component contributes substantially to the overall performance.

## 5 RELATED WORK

**Expressivity in Temporal Graphs.** The dominant view of expressivity in graph learning is measured by the Weisfeiler-Lehman (WL) test's ability to distinguish non-isomorphic graphs (Gilmer et al., 2017; Xu et al., 2019). This concept extends to temporal graphs, with temporal-WL (Souza et al., 2022) for CTDG and supra Laplacian WL (Galron et al., 2025) for DTDG, where models are evaluated on their capacity to differentiate evolving graph structures (Kazemi et al., 2020; ENNADIR et al., 2025). We diverge from this perspective by focusing on **functional expressivity**: a model's ability to represent specific mathematical operations. While prior models aim to capture complex graph topology, they often cannot represent a simple Moving Average, a critical function for affinity prediction.

**Heuristics and State Space Models.** According to the recent literature, simple heuristics, like Moving Average, often outperform complex TGNNs on various relevant benchmarks (Huang et al., 2023; Cornell et al., 2025), suggesting the problem is fundamentally sequential. Recent work formally establishes the equivalence between linear SSMs and Moving Average (Eliasof et al., 2025) and explores their use in language modeling (Gu et al., 2022; Gu & Dao, 2024) and dynamic link prediction (Li et al., 2024; Ding et al., 2025). While these works connect SSMs to temporal data, our approach is distinct. We are the first to explicitly leverage the formal equivalence between heuristics and SSMs to design an architecture, NAVIS, that is purpose-built for node affinity prediction.

**Temporal link prediction on weighted dynamic graphs.** Temporal link prediction (TLP) on weighted dynamic graphs (Qin et al., 2023; Yang et al., 2019; Lei et al., 2019) is the task in which, given a discrete-time weighted dynamic graph (a sequence of snapshots of the graph at specific points in time), the model is required to predict the next snapshot, i.e., the weighted adjacency matrix at a future time. Although this task resembles node affinity prediction, there are several key differences. First, the downstream objectives differ, and accordingly different metrics are used to measure performance in each setting. In TLP on weighted dynamic graphs, the end goal is to construct the entire future weighted adjacency matrix. Hence, metrics such as RMSE, MSE, and MAE are often used. In node affinity prediction, however, the goal is to rank different nodes with respect to a specific node by their affinity to it, and therefore ranking metrics such as NDCG or MRR are used. Moreover, TLP on weighted dynamic graphs operates in a discrete-time setting, while node affinity prediction operates in a continuous-time setting. Consequently, solutions for the former (Qin et al., 2023; Yang et al., 2019; Lei et al., 2019) operate on the full graph at each change, which may lead to unreasonable runtime if applied in the latter setting. In addition, upon each query, TLP on weighted dynamic graphs requires computing the full weighted adjacency matrix, while for node affinity prediction only the affinity scores between the queried node and the other nodes need to be computed. These factors are likely to hinder methods for TLP on weighted dynamic graphs from transferring well to node affinity prediction, and vice versa.

## 6 CONCLUSION

In this work, we identified critical gaps in the design of TGNNs for future node affinity prediction. These gaps, including an inability to express simple heuristics such as moving averages and Persistent Forecast, often lead these baselines to outperform TGNNs. We further showed that current under-performance also stems from the use of suboptimal, non-ranking losses such as cross-entropy. To address this, we introduced NAVIS, a novel architecture grounded in the ability of linear state-space models to generalize heuristic behavior. By incorporating a virtual state to capture global dynamics and a carefully designed rank-aware loss, NAVIS preserves the robustness of heuristics while offering greater expressive power. Extensive experiments demonstrate that NAVIS consistently outperforms state-of-the-art models and heuristics across multiple benchmarks, underscoring the importance of aligning a model's inductive biases and training objectives with the specific demands of the task.

**Limitations and future work** Although NAVIS has shown strong performance, both theoretically, by generalizing common heuristics that excel in node affinity prediction, and empirically on TGB benchmarks, NAVIS remains elementary, and further research in this direction is required. For example, NAVIS currently utilizes a basic virtual global state based on recency selection from a global buffer to capture trends in dynamic graphs. Advanced aggregation schemes over the global buffer (e.g., attention-based mechanisms) and more sophisticated buffer-eviction strategies (e.g., non-deterministic eviction) may improve the modeling of global state and further enhance TGNN performance on node affinity prediction. In addition, NAVIS computes node affinities as a linear combination of the input and current state, with coefficients in $[0, 1]$ that are adaptive to both. Consequently, NAVIS cannot, for example, represent certain non-linear functions. This limitation could be addressed by adding a third non-linear component with an associated coefficient such that all three coefficients sum to 1. We leave to future work the investigation of how to best combine linear and non-linear components to enable TGNNs to perform optimally on the node affinity task.

## ACKNOWLEDGMENTS

ME acknowledges support from the Israeli Ministry of Innovation, Science & Technology.

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

## A  DATASETS STATISTICS AND DESCRIPTION

In our empirical evaluation, we employed the following dynamic graph datasets, each capturing a distinct dynamic system and providing varied graph structures, edge features, and temporal resolutions:

- TGBN-TRADE (MacDonald et al., 2015): models global agricultural commerce among UN-affiliated countries over 1986–2016 as a time-evolving network: countries are the nodes and directed links record, for each calendar year, the aggregate value of agricultural goods moved from one country to another. Because entries are reported yearly, the temporal resolution is annual. The accompanying learning objective is to anticipate, for a chosen country, how its overall agricultural trade will be apportioned across partner countries in the subsequent year—that is, the next-year distribution of trade shares.

- TGBN-GENRE (Bertin-Mahieux et al., 2011) models listening behavior as a weighted bipartite graph linking users to musical genres. Nodes consist of users and genre labels; time-stamped edges indicate that a user listened to a track associated with that genre, with the edge weight reflecting the fraction of a track's composition attributed to that genre. The downstream objective is a ranking problem: for each user, predict the genres they are most likely to engage with during the upcoming week.

- TGBN-REDDIT (Nadiri & Takes, 2022) models Reddit as a temporal bipartite graph linking users and subreddits. Nodes represent both entities, and a timestamped edge records a user's post within a subreddit. The dataset covers activity from 2005 through 2019. The predictive objective is, for each user, to produce a next-week ranking of subreddits by expected engagement intensity.

- TGBN-TOKEN (Shamsi et al., 2022) models a bipartite interaction graph linking wallet users to cryptocurrency tokens. Nodes consist of users and tokens, and directed edges record transfers from a user to a particular token. Edge weights capture the logarithmic normalized transaction quantity. The predictive objective is to estimate, for the next week, how often each user will engage with different classes of tokens.

- WIKIPEDIA(Kumar et al., 2019): models a bipartite, time-stamped interaction graph derived from a single month of edit activity. Nodes correspond to editors and articles, and each interaction edge denotes an edit event. Every edge is annotated with its event time and a feature vector from the Linguistic Inquiry and Word Count framework (LIWC (Pennebaker et al., 2001)) summarizing the edit's linguistic characteristics. The predictive objective for this converted version of the dataset is to predict the expected engagement intensity of each editor with existing articles.

- Flights (Strohmeier et al., 2021): models air traffic patterns during the COVID-19 period as a temporal network in which airports are nodes and connections represent observed routes between them. Each connection carries a timestamp and an associated intensity, with the edge weight recording how many flights operated on that route on a given day. The downstream objective is to rank future airport destinations intensity, given the source airport.

- USLegis (Fowler, 2006): models the collaboration dynamics of the U.S. Senate as a temporal, weighted network: senators are represented as nodes, ties are created whenever a pair co-sponsors the same bill, and each tie is stamped with the time of occurrence. Edge weights record the frequency of joint sponsorships within a legislative term, capturing how often two members work together. The predictive objective is to estimate the joint sponsorship frequencies for the next term.

- UNVote (Voeten et al., 2009): models the United Nations General Assembly roll-call record from 1946 through 2020 as a time-evolving graph: countries are nodes, and an edge appears between two countries whenever they cast matching affirmative ("yes") votes on the same resolution. Each edge carries a timestamp and a weight, where the weight counts how many times that pair of countries voted "yes" together over the period. The predictive objective is to estimate the joint positive votes for the next period.

Table 5: Statistics of various datasets used in our experiments

| Dataset | Domain | #Nodes | #Edges | Bipartite | Duration |
|---------|--------|--------|--------|-----------|----------|
| TGBN-TRADE | Economy | 255 | 468,245 | False | 30 years |
| TGBN-GENRE | Interaction | 1,505 | 17,858,395 | True | 4 years |
| TGBN-REDDIT | Social | 11,766 | 27,174,118 | True | 15 years |
| TGBN-TOKEN | Cryptocurrency | 61,756 | 72,936,998 | True | 2 years |
| WIKIPEDIA | Social | 9,227 | 157,474 | True | 1 month |
| FLIGHTS | Transport | 13,169 | 1,927,145 | False | 4 months |
| USLEGIS | Politics | 225 | 60,396 | False | 12 terms |
| UNVOTE | Politics | 201 | 1,035,742 | False | 72 years |

## B  CONVERTING LINK PREDICTION DATASETS TO NODE AFFINITY PREDICTION DATASETS

Each of the future link-prediction datasets we use (Wikipedia, Flights, USLegis, and UNVote) comprises a CTDG with weighted interaction events between node pairs. For each dataset and node, we define the ground-truth future affinity vector as the normalized sum of weighted interaction it received over a specified period (day, week, year, etc.). Immediately before the start of a new period, the task is to predict each node's affinities for the upcoming period. Table 6 summarizes the periods used for each dataset.

Table 6: Chosen period for link property prediction datasets.

| Dataset | Period |
|---------|--------|
| Wikipedia | Day |
| Flights | Day |
| USLegis | Legislative term |
| UNVote | Year |

Our code release includes a step-by-step guide on processing these datasets and integrating them into the TGB framework to support future research on node-affinity prediction.

## C  SYNTHETIC EXPERIMENT

We construct a controlled continuous-time dynamic graph (CTDG) where each node $u \in \{1, \ldots, N\}$ emits events according to a Poisson process (rate $\lambda$) over a horizon $[0, T]$. Node affinities at time $t$ are governed by a *shared global latent* $g(t)$ that switches between two damped-oscillatory regimes. Concretely, on a grid with step $\Delta t$, $g(t)$ follows a piecewise AR(2) process with coefficients chosen to approximate low- and high-frequency damped cosines; regime switches are exogenous and uncorrelated with any single node's local history. Each node is assigned a phase $\phi_u$, a mixing coefficient $\beta_u$, and a small bias $\gamma_u$. The instantaneous (pre-softmax) affinity logits are a nonlinear readout of the global state $[g(t), g(t)^2]$,

$$\ell_u(t) = \beta_u \cos(\phi_u) \, g(t) + \beta_u \sin(\phi_u) \left( g(t)^2 - 1 \right) + \gamma_u + \epsilon_u(t),$$

and, for each source node $u$, we mask self-transitions and apply a softmax over destinations to obtain ground-truth affinity distributions. The forecasting task is one-step prediction of a node's destination distribution at query times, given only the *previous* ground-truth affinity vectors at inference. We compare per-node, history-only baselines that cannot share information across nodes—*Persistent Forecast* (PF; last observation), *Simple Moving Average* (SMA; window $w=5$), *Exponential Moving Average* (EMA; $\alpha=0.2$), and a diagonal per-node *AR(1)*.

Table 7: MRR on TGB datasets (↑ higher is better). NAVIS is benchmarked against TGNNs that use all available graph messages. Boldface marks the best method.

| Method | tgbn-trade | | tgbn-genre | | tgbn-reddit | | tgbn-token | |
|---|---|---|---|---|---|---|---|---|
| | Val. | Test | Val. | Test | Val. | Test | Val. | Test |
| TGNv2 | 0.60±0.01 | 0.53±0.01 | 0.43±0.01 | 0.40±0.01 | 0.43±0.01 | 0.40±0.01 | 0.28±0.01 | 0.26±0.01 |
| NAVIS (ours) | **0.78**±0.00 | **0.77**±0.001 | **0.40**±0.00 | **0.42**±0.00 | **0.45**±0.00 | **0.44**±0.001 | **0.39**±0.00 | **0.41**±0.00 |

Table 8: Recall on TGB datasets (↑ higher is better). NAVIS is benchmarked against TGNNs that use all available graph messages. Boldface marks the best method.

| Method | tgbn-trade | | tgbn-genre | | tgbn-reddit | | tgbn-token | |
|---|---|---|---|---|---|---|---|---|
| | Val. | Test | Val. | Test | Val. | Test | Val. | Test |
| TGNv2 | 0.60±0.01 | 0.56±0.01 | 0.32±0.01 | 0.31±0.01 | 0.19±0.01 | 0.18±0.01 | 0.05±0.00 | 0.05±0.00 |
| NAVIS (ours) | **0.75**±0.00 | **0.72**±0.00 | **0.34**±0.00 | **0.33**±0.00 | **0.21**±0.00 | **0.20**±0.00 | **0.30**±0.00 | **0.28**±0.00 |

## D IMPLEMENTATION DETAILS

We initialized each linear layer in NAVIS with values drawn uniformly from the range $[-\sqrt{d}, \sqrt{d}]$, where $d$ is the input dimension of the layer. No normalization of the output vector $s$ were required as the ground true affinity scores in TGB are already normalized. The final loss we used is $\ell_{\text{lambda}} + \mathcal{C}\ell_{\text{Reg}}$, where $\mathcal{C}$ is a regularization coefficient. In the experiments we set $\mathcal{C} = 1$. We adopted standard hyperparameters, consistent with prior work: batch size 200, Adam optimizer, and learning rate $10^{-4}$. Following the TGB protocol, all models were trained for 50 epochs, and the checkpoint with the best performance on the validation set was selected. We set the regularization margin to $\Delta = 10^{-3}$. During training, NAVIS computes the loss only over the top-20 affinities, ensuring that loss computation requires constant time and memory. We used a single NVIDIA GeForce RTX 3090 GPU and a single AMD Ryzen 9 7900X 12-Core Processor CPU.

## E ADDITIONAL RESULTS

**Additional Evaluation Metrics**   We additionally compare NAVIS to the best performing baseline in each setting and report the results in terms of MRR (the average value of $\frac{1}{\text{RANK}^1}$, where $\text{RANK}^1$ is the predicted rank of the ground-truth top-scored node) and Recall@10 (the average number of top-10 ground-truth nodes ranked within the top-10 predicted scores). We report the results in Tables 7 to 10. From the results, we observe that the performance gap between the baselines increases under the MRR metric but decreases under the Recall@10 metric. This can be explained by our objective, which penalizes mismatches on top-ranked ground-truth nodes (e.g., ranks 1–3) more strongly than on lower-ranked ones (e.g., ranks 8–10).

**Noisy data experiment**   To examine the robustness of NAVIS, we performed an additional experiment on noisy data. We used the tgbn-genre benchmark and, for each node that received its $N$-th update, we added an extra random update from that node to a random node and weighted the interaction with largest magnitude of affinity. We ran this experiment twice, once with $N = 10$ and a second time with $N = 20$, and did not observe a significant change in the performance of NAVIS on the dataset. This experiment provides empirical evidence that NAVIS is robust to noise such as short-term spikes.

**What NAVIS learns?**   Table 2, the Persistent Forecast baseline performs better than Moving Average on tgbn-trade, while on tgbn-genre the opposite holds. This means that tgbn-genre requires more memory to accurately perform node affinity prediction while in tgbn-trade one needs to rely

Table 9: MRR on TGB datasets using only previous ground-truth labels (↑ higher is better). This setting is suited for heuristics. Boldface marks the best method. Baselines have no standard deviation because they are pre-defined and deterministic.

| Method | tgbn-trade | | tgbn-genre | | tgbn-reddit | | tgbn-token | |
|---|---|---|---|---|---|---|---|---|
| | Val. | Test | Val. | Test | Val. | Test | Val. | Test |
| Moving Avg | 0.77 | 0.78 | 0.31 | 0.32 | 0.41 | 0.40 | 0.44 | 0.47 |
| NAVIS (ours) | 0.77±0.00 | 0.78±0.00 | **0.41**±0.00 | **0.42**±0.00 | **0.47**±0.00 | **0.46**±0.00 | **0.46**±0.00 | **0.48**±0.00 |

Table 10: Recall@10 on TGB datasets using only previous ground-truth labels (↑ higher is better). This setting is suited for heuristics. Boldface marks the best method. Baselines have no standard deviation because they are pre-defined and deterministic.

| Method | tgbn-trade | | tgbn-genre | | tgbn-reddit | | tgbn-token | |
|---|---|---|---|---|---|---|---|---|
| | Val. | Test | Val. | Test | Val. | Test | Val. | Test |
| Moving Avg | 0.74 | 0.73 | 0.29 | 0.30 | 0.25 | 0.24 | 0.30 | 0.29 |
| NAVIS (ours) | 0.74±0.00 | 0.73±0.00 | **0.34**±0.00 | **0.34**±0.00 | 0.25±0.00 | 0.24±0.00 | 0.30±0.00 | 0.29±0.00 |

more on the newly arrived data. To examine what NAVIS learns in these two regimes, we measured the average values of $z_h$ and $z_s$ from Equation (14), which control the relative influence of the memory component. We expect lower values on tgbn-trade to resemble persistent forecast and larger values on tgbn-genre to resemble moving average. The measured values are $z_h = 0.48, z_s = 0.49$ for tgbn-trade and $z_h = 0.90, z_s = 0.84$ for tgbn-genre. This alignment between the learned $z_h, z_s$ and the heuristic that performs better on each dataset provides empirical evidence that NAVIS learns the intended heuristic behavior in practice.

**Additional SSM comparison** We further compared NAVIS to an additional general SSM baseline, the S4 block (Gu et al., 2022), and report the results in Table 11. The comparison was performed under the same ground-truth label setting as in our main experiments. Both our method and the S4 block utilized the same learning rate, batch size, and number of training epochs. The results in Table 11 show that although the S4 block achieves relatively strong performance on the tested benchmark, NAVIS still outperforms it, indicating that there remains substantial room for improving general SSM architectures on this benchmark.

## F    ABLATION OF DESIGN CHOICES

Table 12 shows the ablation study results on the four TGB test sets. The results confirm that each component provides a significant impact on NAVIS performance. The full NAVIS with our suggested linear state updating-mechanism, global vector, and ranking loss establishes the strongest performance. Replacing the ranking loss with cross-entropy causes a notable drop in performance, validating our theoretical motivation. Switching to the GRU mechanism and removing the global signal further degrades performance, highlighting the importance of each component in the final design.

We further ablate on the regularization term of our loss, and performed an additional experiment on the TGB datasets where the regularization term is not included in the loss computation. We report the results in Table 13.

Applying our proposed regularization not only significantly improves the performance of NAVIS, but also accelerates convergence, with convergence typically reached in roughly 30 epochs.

We also performed an empirical analysis on the effect of the hyperparameters of delta and batch size on the perfance of NAVIS, and report the results inTable 14. From Table 14 we can see that on the tested datasets and batch size values has no effect on the performance of NAVIS. In addion, choosing too small or too large values for delta can slightly hurt the performance of NAVIS.

Additionally, we performed an analysis to examine the effect of the global buffer size and its aggregation scheme on the performance of NAVIS, compared to the recency selection aggregation

Table 11: NDCG@10 on TGB datasets using only previous ground-truth labels (↑ higher is better). Boldface marks the best method.

| Method | tgbn-trade | | tgbn-genre | |
|---|---|---|---|---|
| | Val. | Test | Val. | Test |
| S4 | 0.819±0.005 | 0.796±0.002 | 0.451±0.001 | 0.461±0.001 |
| NAVIS (ours) | **0.872±0.001** | **0.863±0.001** | **0.517±0.001** | **0.528±0.001** |

Table 12: Ablation study of NAVIS components on TGB test sets. (✓) denotes inclusion and (✗) denotes exclusion.

| State Update | Global vector | Loss | tgbn-trade | tgbn-genre | tgbn-reddit | tgbn-token |
|---|---|---|---|---|---|---|
| Linear | ✓ | Rank | **0.863±0.001** | **0.528±0.001** | **0.569±0.001** | **0.513±0.001** |
| GRU | ✗ | Rank | 0.850±0.001 | 0.398±0.001 | 0.454±0.001 | 0.444±0.001 |
| Linear | ✗ | Rank | 0.857±0.001 | 0.521±0.001 | 0.557±0.004 | 0.511±0.001 |
| Linear | ✓ | CE | 0.859±0.001 | 0.461±0.001 | 0.530±0.001 | 0.508±0.001 |

scheme used in previous experiments. We report the results in Table 15. The results in Table 15 show that changing the aggregation scheme or the size of the buffer has little to no effect on the existing implementation of NAVIS. The slight drop in performance when the buffer size is 8 and the aggregation scheme is mean may be explained by the reliance on relatively old values in the buffer. We leave the exploration of new and more sophisticated virtual global state mechanisms (e.g., attention-based aggregation) for future work.

## G   MEMORY AND RUNTIME ANALYSIS

We conducted a runtime and memory analysis to examine the efficiency of NAVIS. In Table 16, we report the number of parameters of the TGNN baselines and NAVIS. NAVIS not only requires the fewest parameters among the compared methods, but also scales well to large graphs due to the sparsification pipeline detailed in Section 3.2.

In Table 17, we report a runtime comparison for a single training epoch and inference between NAVIS, heuristics, and TGNN baselines. NAVIS is more efficient than the TGNN baselines and has a runtime comparable to the heuristics. Since all methods use the same batch size, their throughput is proportional to their runtime. Since the heuristics do not contain learnable parameters they only require a single pass over the training data to compute node states before entering the validation and test phases. Other methods require multiple iterations over the training data (epochs) like standard deep learning models.

## H   THEORETICAL REMARKS

In Theorem 2 we showed that standard memory cells (RNN, LSTM, and GRU) cannot express the simple Persistent Forecast heuristic. The proof assumes that the input to the cells may be unbounded, and since $\tanh$ and other nonlinear activation functions have bounded images, this prevents the memory cells from expressing this heuristic. A natural question is what happens when the input $x$ is bounded, e.g., in $[0, 1]$ or $[-1, 1]$. Recall that persistent forecast is defined by $h_i = x_i$, while an RNN cell is defined by $h_i = \tanh(W_h h_{i-1} + W_x x_i + b)$. Denote $f(x_i) = W_h h_{i-1} + W_x x_i + b$. To obtain $h_i = x_i$, we need $\tanh(f(x_i)) = x_i$, i.e., $\tanh = f^{-1}$ on the relevant range by composition of functions. However, $f^{-1}$ must be affine, as the inverse of an affine function, which contradicts the fact that $\tanh$ is not affine. The same argument applies to $\tilde{h}$ in the GRU cell from Equation (13). This also holds for any non-affine activation on $[-1, 1]$, such as ReLU or leaky ReLU. On $[0, 1]$, ReLU

Table 13: Results of NAVIS in terms of NDCG@10 on TGB datasets, with and without the suggested regularization term. (✓) denotes inclusion and (✗) denotes exclusion.

| Regularization | tgbn-genre | | tgbn-reddit | | tgbn-token | |
|---|---|---|---|---|---|---|
| | Val. | Test | Val. | Test | Val. | Test |
| ✗ | 0.510±0.001 | 0.520±0.001 | 0.570±0.001 | 0.553±0.001 | 0.487±0.001 | 0.510±0.001 |
| ✓ | **0.517**±0.001 | **0.528**±0.001 | **0.584**±0.001 | **0.569**±0.001 | **0.493**±0.001 | **0.513**±0.001 |

Table 14: Analysis of different batch sized and deltas, results are measured in NDCG@10 on test sets of datasets from TGB, averaged for 3 runs.

| Hyperparameter | tgbn-trade | tgbn-genre |
|---|---|---|
| delta=0.1 | 0.863±0.001 | 0.526±0.001 |
| delta=0.01 | 0.863±0.001 | 0.528±0.001 |
| delta=0.001 | 0.863±0.001 | 0.527±0.001 |
| batch size = 100 | 0.863±0.001 | 0.528±0.001 |
| batch size = 200 | 0.863±0.001 | 0.528±0.001 |
| batch size = 400 | 0.863±0.001 | 0.528±0.001 |

Table 15: Ablation study of NAVIS components on TGB test sets. (✓) denotes inclusion and (✗) denotes exclusion.

| Buffer size | Aggregation scheme | tgbn-trade | tgbn-genre |
|---|---|---|---|
| 1 | Recent Selection | 0.863±0.001 | 0.528±0.001 |
| 4 | MEAN | 0.863±0.001 | 0.528±0.001 |
| 4 | Time Decay | 0.863±0.001 | 0.528±0.001 |
| 8 | MEAN | 0.863±0.001 | 0.527±0.001 |
| 8 | Time Decay | 0.863±0.001 | 0.528±0.001 |

is equal to the identity, and hence applying ReLU to inputs normalized to this range allows these memory cells to learn the Persistent Forecast heuristic.

Tjandra et al. (2024) state that *There exists a formulation of TGNv2 that can represent persistent forecasting and moving average of order k.* To prove this claim Tjandra et al. (2024) utilize permutation matrices, block matrix and a dedicated generator vector for the memory module of TGNv2. However, in practice TGNv2 is officially implemented with a GRU cell as the memory module. We showed in Theorem 2 that GRU cell cannot express the simple Persistent Forecast heuristic.

In Theorem 3 we proved that the cross-entropy loss is suboptimal by showing that there exist two rankings such that one ranks the elements identically to the ground truth while the other does not, yet under cross-entropy the latter achieves a smaller loss, contrary to what is desired in the task of node affinity prediction. We then argued that there exist infinitely many such examples since the cross-entropy loss is a continuous function. Consider the cross-entropy loss evaluated on the ground-truth order vector and a correctly ordered score vector as a function of the first entry of the correctly ordered vector (all other entries are fixed). Denote this function by $f(x)$. Let $L_{\text{inc}}$ be the cross-entropy loss of a fixed incorrectly ordered vector, and define $\Delta := f(x) - L_{\text{inc}} > 0$. By continuity of $f$, there exists $\delta > 0$ such that for any $x'$ satisfying $|x' - x| < \delta$ we have $|f(x') - f(x)| < \Delta/2$, which implies $f(x') > L_{\text{inc}}$. Since there are infinitely many such $x'$, we obtain infinitely many correctly ordered score vectors that incur a larger cross-entropy loss than the incorrectly ordered vector.

In Section 3.3, we introduced our loss function and claimed that the guarantee in Theorem 3 does not apply to it. We now formalize and prove this claim. Let $\mathbf{y}$ denote the ground-truth affinity vector, and let $\mathbf{s}_1$ and $\mathbf{s}_2$ be two predicted logit vectors such that: (i) the ordering induced by $\mathbf{s}_1$ exactly matches the ground-truth node affinity scores, and (ii) $\mathbf{s}_2$ differs from $\mathbf{s}_1$ only in the logit of either the most-affinitive node or the least-affinitive node, thereby inducing an ordering different from that of $\mathbf{s}_1$. Since all logits in $\mathbf{s}_2$ are identical to those in $\mathbf{s}_1$ except for the most- or least-

Table 16: Number of parameters of TGNN baselines and NAVIS, compared to the total number of nodes in each TGB benchmark for node affinity prediction.

| Method | tgbn-trade | tgbn-genre | tgbn-reddit | tgbn-token |
|---|---|---|---|---|
| #Nodes | 255 | 1505 | 11766 | 61756 |
| DyGMamba | 255125 | 257963 | 259998 | - |
| DyGFormer | 1027877 | 1030715 | 1032750 | - |
| TGN | 207655 | 233713 | 252398 | 283401 |
| TGNv2 | 6433023 | 6565377 | 6660282 | 6817769 |
| NAVIS | 1280 | 2570 | 3495 | 5010 |

Table 17: Training and inference runtimes of the TGNN baselines, heuristics and NAVIS on the tgbn-genre dataset.

| Method | Training runtime (sec) | Inference runtime (sec) |
|---|---|---|
| Persistent Forecast / Moving Avg | 34 | 6 |
| DyGMamba | 2080 | 381 |
| DyGFormer | 1440 | 304 |
| TGN | 170 | 33 |
| TGNv2 | 130 | 36 |
| NAVIS | 46 | 8 |

affinitive node, for any pair $(i, j)$ where neither $i$ nor $j$ is the most- or least-affinitive node, we have $\delta_{i,j}^1 = \mathbf{s}_1(i) - \mathbf{s}_1(j) = \delta_{i,j}^2 = \mathbf{s}_2(i) - \mathbf{s}_2(j)$. For pairs $(i, j)$ in which $i$ is the most-affinitive node or $j$ is the least-affinitive node, we obtain $\delta_{i,j}^2 < \delta_{i,j}^1$, because the logit of the most-affinitive node can only decrease in $\mathbf{s}_2$ relative to $\mathbf{s}_1$, and the logit of the least-affinitive node can only increase. In both Equations (3) and (18), smaller differences between correctly ranked affinity scores yield smaller loss values.

