# OpenReview forum: "Revisting Node Affinity Prediction In Temporal Graphs"
_ICLR.cc/2026/Conference — ICLR 2026 Poster_

### Official Review · Reviewer_QArm · 2025-10-26

**Soundness:** 2
**Presentation:** 3
**Contribution:** 3
**Rating:** 6
**Confidence:** 3

**Summary:**

This work considers the node affinity prediction on continuous-time dynamic graphs (CTDGs), a more challenging yet realistic task than the conevntional temporal link prediction. Based on some simple analysis regarding the (i) equivalance between heuristic and state space models as well as (ii) failure of widely-used cross-entroy loss, a new NAVIS (Node Affinity prediction model using Virtual State) method was then proposed, with some original designs of global virtual state and using regularized lambda loss. Experiments on large-scale dyanmic benchmark (i.e., TGB) validates the effectiveness of NAVIS.

**Strengths:**

**S1**. The overall presentation of this paper is well-written and well-motivated, and thus easy to read.

**S2**. As claimed in the paper, node affinity prediction is a more chellenging yet realistic task compared with the conventional temporal link prediction.

**S3**. The proposed method was evaluted on large-scale dyanmic graph benchmark (i.e., TGB)

**S4**. This work anonymously provide its code to ensure the reproducibility of experiments.

**Weaknesses:**

**W1**. There are no discussions about some related work about the inference on weigthed dynamic graphs.

According to the problem statement in Section 2, the proposed method considers weighted dynamic graphs, which each edge assocaited with a weight in addition to a timestamp. Different from conventional temporal link prediction (TLP) on unweighted graphs, there are some prior studies [1-3] consider TLP on weigthed dynamic topology (although they still adopt the data model of discret-time dynamic graphs), which are not discussed in this work. In particular, [1-3] can effectively tackle the wide-value-range and sparsity issues of weigthed dynamic graphs. Can the proposed method handle these issues?

[1] GCN-GAN: A Non-linear Temporal Link Prediction Model for Weighted Dynamic Networks. InfoCom 2019.

[2] An Advanced Deep Generative Framework for Temporal Link Prediction in Dynamic Networks. IEEE Transactions on Cybernetics 2020.

[3] High-Quality Temporal Link Prediction for Weighted Dynamic Graphs via Inductive Embedding Aggregation. TKDE 2023.


***
**W2**. From the perspective of rigirous theoretical analysis, Theorem 1 and Theorem 2 are more likely to be facts rather than theorems, as they are staightforward. For Eq. (11), setting $\phi$ to tanh is just one possible choice of the non-linear activation function. In addition, one can also set $\phi$ to ReLU, LeakyReLU, etc. Will these settings change the main results of Theorem 2? It seems that the issue mentioned in Theorem 2 can be easily handled by first normalizing the input into [0, 1] or [-1, 1] and finally recovering to the original value range for output. Will this simple modification chanege the main results of Theorem 2?

It seems that Theorem 3 was proved only based on a specific toy example, which may not be a rigrious proof to show that there exist infinitely many triplets.


***
**W3**. Some details about the proposed method are missing.

After reading Section 3.2, it remains unclear how to derive a feasble prediction result (e.g., normalized ${\bf \hat x}$) using the new structure defined in (14), as there is no ${\bf \hat x}$ in Eq. (14).

There are no formal equations to explictly describe how to use Eq. (14) to derive feasible outputs for a batch of nodes.

It seems that the final training loss of NAVIS was not foramlly given in tha main paper, which may be a combination of Eq. (16) and (18) with a tunable hyper-parameter.

There is no pseudo-code to summarize the overall training and inference procedures of NAVIS, in which some of the aforemention details can be checked.


***
**W4**. Current experiment setups may not fully validate the effectiveness of NAVIS. Some more further experiments are suggested.

The pre-experiments shown in Fig. 1 were evaluated based on MAE and mean rank, while main experiment results in Tables 1-4 were based on NDCG@10, which may not be consistent.

There seem no parameter analysis to test the effects w.r.t. different settings of hyper-paramters.

There seems no comparison about the training time, inference time, and memory consumption. As NAVIS may potentially result in a better trade-off between the inference quality and effciency of node affinity prediction, some further analysis about its efficiency is reommended.

Table 5 does not provide details about the value range of edge weights. As mentioned in **W1**, wide-value-range issue may make the inference on weigthed graphs more challenging than that on unweighted topology.


***
**W5**. There are no discussion about the limitations of this work and possilbe solutions as future research directions.

**Questions:**

See **W1**-**W5**.

---

> ### Author Response · Authors · 2025-11-20
> **Part 1 of comment for Reviewer QArm**
>
> Dear Reviewer QArm, we are delighted to read that you find the problem discussed in the paper valuable, and that you find the manuscript clear. We are also thankful for the constructive feedback, which we address in our responses below. We have also revised our paper accordingly, with changes marked in blue.
>
> 1. **Regarding related work:** Thank you for the insightful comment. We agree that TLP on weighted dynamic graphs is closely related to node affinity prediction, and we appreciate your provision of three interesting and related works [1–3]. To address your comment, we have now added these works and a dedicated paragraph on TLP over weighted dynamic graphs in the Related Work section. We also note that there are several differences that require dedicated solutions for each task. To be concrete: *First*, as mentioned in your review, existing TLP methods on weighted dynamic graphs are defined in a discrete-time setting, whereas node affinity prediction problem is formulated in continuous time. Methods like  [1–3] operate on the full graph at each discrete time step, which would lead to prohibitive runtime if applied at the granularity of every event in a continuous-time setting (e.g., tgbn-token has 72,936,998 updates). *Second*, upon each query, methods for TLP on weighted dynamic graphs usually reconstruct the full weighted adjacency matrix, which does not scale well when one is interested in efficient, per-node queries as in node affinity prediction. *In contrast*,  NAVIS and other methods for node affinity precision are designed to answer node affinity queries directly without reconstructing the entire adjacency matrix, making it better aligned with the literature [1,2]. Third, the performance of  TLP methods on weighted dynamic graphs are often measured by their reconstruction error of the weighted adjacency matrix, and therefore they are optimized accordingly. In contrast, in the node affinity prediction task, the performance of the models is measured based on the predicted ranking of the node by their affinities, rather than accurately computing the ground truth affinities. Regarding the wide-value-range and sparsity issue. As mentioned in Appendix D, because the task of node affinity is to predict and order the top neighbors with the highest affinities, NAVIS computes the loss only over the top-20 affinities, ensuring that loss computation requires constant time and memory regardless of the sparsity level of the graph. In light of your comments, we have revised the paper to include these important discussions in the related work section. Thank you for helping us improve the work and better position our contribution.
>
>
> 2.**Regarding analysis:** We embrace your constructive feedback. Regarding Theorem 2, we  assumed unbounded data to simplify the proof. In fact, RNN cells and GRU cells cannot reproduce persistent forecast even when the data is normalized. Assume that $x_i$​ is normalized such that $x_i \in [0,1]^d$. An RNN cell is defined as $h_i = tanh(W_h h_{i-1} + W_x x_i + b)$. Reproducing PF means $h_i = x_i$ for all possible sequences $x_i \in [0,1]^d$. Mark $f(x_i) = W_h h_{i-1} + W_x x_i + b$. Here $f$ is an affine (linear + bias) function. To satisfy $h_i = x_i$ for all $x_i \in [0,1]^d$, we would need $f(x_i) = tanh^{-1}(x_i)$ on $[0,1]^d$, but $tanh^{-1}$ is strictly non-linear on [0,1], so an affine function cannot coincide with it on the whole interval. Hence, no choice of  $Wh$,$Wx$,$b$ can reproduce persistent forecast for all normalized inputs. The same argument applies to $\tilde{h}_i$​ from Eq. (13) (the GRU cell), where the update is again obtained by applying a non-linear activation to an affine function. This reasoning shows that the impossibility holds for any activation that is non-linear. For the specific activations you mention: on [0,1], ReLU and LeakyReLU reduce to the identity function, so the above argument does not apply in that restricted setting (they are no longer strictly non-linear there). When the data is normalized in [−1,1], however, ReLU is non-linear on the interval (it clips all negative values to 0), and the same impossibility argument applies. Since the TGB datasets we use are already normalized [1], this actually strengthens our claim in the regime we study. We have clarified the precise assumptions on the activation function and added the discussed remarks in Appendix H of the revised manuscript. Regarding Theorem 3, the cross-entropy loss is a continuous function, and the inequality between the two candidate solutions in our example is strict. Therefore, by continuity, there exists an open neighborhood of inputs obtained by arbitrarily small perturbations of this example such that the same inequality of cross-entropy values holds, which implies that there are in fact infinitely many such triplets. We have now stated this argument explicitly and formally in Appendix H. Thank you.

---

> > ### Author Response · Authors · 2025-11-20
> > **Part 2 of comment for Reviewer QArm**
> >
> > 3.**Regarding method details:** Thank you for the comment. The vector $s$ in Eq. (14) is the output of NAVIS: each entry in $s$ represents the affinity between the queried node and other nodes in the graph. These affinity scores are directly used as the prediction outputs of NAVIS. No additional normalization step is required. To apply NAVIS to a batch of nodes, Eq. (14) is simply evaluated in parallel for all nodes in the batch. As correctly noted in your review, the loss used to train NAVIS is a combination of Eq. (16) and Eq. (18) with a tunable hyperparameter. We have made this explicit in the revised manuscript in Appendix D. Our original submission (as well as the revised one) provide the full NAVIS code to further ease understanding of the implementation and prediction pipeline. We thank you for the feedback that helped us improve the clarity of our work.
> >
> > 4.**Regarding experiments setups:** Thank you for the comment. To accommodate it, we revised Appendix E, Appendix F and Appendix G, and we now include additional experiments addressing these points: *(1)* We now report results in terms of MRR and Recall, in addition to NDCG@10, and we observe that the relative performance of NAVIS and the baselines is consistent across all these metrics, *(2)* We conduct a parameter analysis with respect to key hyperparameters of NAVIS, and *(3)* We compare the number of parameters, training time, inference time, and discuss the corresponding memory consumption, highlighting the efficiency of NAVIS. Finally, in all datasets the edge weights (affinities) are normalized officially by TGB [1]  to be in [0,1], which clarifies the value range considered in Table 5. Thank you.
> >
> > 5.**Regarding limitations:** To address your important comment, we added a dedicated “Limitations and Future Work” section to the revised paper, where we discuss the main constraints of our approach and outline several promising research directions. Thank you.
> >
> > We would like to express our gratitude for your thoughtful and constructive feedback on our paper. We sincerely believe that the additional discussions and experiments that stem from your feedback, helped us to improve our paper. If you have additional concerns or questions, please let us know, and we will be happy to address them. Otherwise, we will appreciate it if you can consider revising your score.
> >
> > **References:**
> >
> > [1] Huang, Shenyang, et al. "Temporal graph benchmark for machine learning on temporal graphs.", 2023
> >
> > [2] Tjandra, Benedict Aaron, Federico Barbero, and Michael Bronstein. "Enhancing the expressivity of temporal graph networks through source-target identification.

---

### Official Review · Reviewer_6w2j · 2025-10-29

**Soundness:** 3
**Presentation:** 3
**Contribution:** 3
**Rating:** 6
**Confidence:** 4

**Summary:**

The authors consider the task of node affinity prediction in temporal networks, which they describe as more challenging than temporal link prediction because it concerns predicting node affinities for a set of possible neighbour nodes at a future time rather than the existence of only one link. They consider simple yet strong baselines for node affinity prediction; specifically, baselines as simple as predicting a recent ground-truth affinity vector for a future point in time, or a moving average over previous affinity vectors. Then, they argue (and show directly) that linear state space models can capture the heuristics defined by those simple baselines. However, they also show that temporal GNNs that use RNN, LSTM, or GRU memory cells suffer from fundamental limitations such that they cannot learn those simple baselines. To address this shortcoming of TGNNs, the authors develop a GNN based on linear state space models for learning node affinities and call it NAViS. On a range of benchmark datasets from TGB, the authors demonstrate that NAViS outperforms current TGNNs and simple heuristic baselines.

I found the paper well written and the authors' arguments easy to follow. I believe the paper makes a valuable contribution, and I found it refreshing to see that the authors considered the possibility that TGNNs suffer from fundamental limitations, which, once discovered, allowed them to formulate a suitable solution. I was also happy to see that the authors highlighted the importance of "aligning model inductive bias and training objectives with the task", which I believe should be more often in focus than it tends to be.

**Strengths:**

- The paper is well written, and the authors' arguments are easy to follow.
- The authors consider a relevant research question and identify fundamental limitations of TGNNs that prevent TGNNs from learning simple heuristics that are strong baselines in node affinity prediction.
- The empirical evaluation, including an ablation study, demonstrates the good performance of the authors' proposed method.

**Weaknesses:**

- I believe there might be a claim that is not fully substantiated. The authors claim that "any memory-based TGNN that applies standard memory cells cannot represent even the simplest heuristics -- persistent forecasting.", which I agree with. However, they state that "[TGNNs] lack the functional expressivity required for node affinity prediction", which I find questionable. The only thing that was established is that they cannot do so by learning such simple baselines. But it has not been established that this is the only way to make good node affinity predictions.
- The authors do not discuss the complexity of their approach, leaving it open whether it scales to large-scale networks.

**Questions:**

I have a couple of questions about points I did not fully understand when reading the paper, and I hope that the authors can help me clarify them.

1. I am unsure whether the main questions from page 1 were answered: It became clear why heuristics outperform more sophisticated TGNNs, but I do not believe that you answered the question of whether we can push TGNNs to do better. But perhaps I have simply missed the answer. Could you elaborate whether your work suggests ways, or even a general recipe, for how to improve TGNNs for node affinity prediction?
2. I am curious whether, as claimed in the first point of the contributions, we can really say that NAViS is more expressive than other TGNNs that use memory cells such as RNN, LSTM, or GRU. I believe that, at least in some regards, NAViS is more expressive because it is designed to be powerful enough to represent what linear state space models can do, and you showed that conventional memory-cell-based TGNNs cannot do that. However, I suspect that there are things that memory-cell-based TGNNs can do that NAViS cannot do---is that correct?
3. Connected to the previous question, is there a way to check whether NAViS actually learns simple heuristics, such as persistent forecasting? Typically, expressivity and learnability are different aspects, and just because NAViS can express something, it doesn't necessarily mean that it can actually learn what is desired in practice. I understand that you used the Lambda Loss specifically to nudge the training in the right direction. Nevertheless, I'm wondering whether you have checked, or whether it is at all possible to check, that NAViS learns what you want it to learn?
4. L.423 mentions that the performance of current TGNNs "[suggests] an incompatible design choice of TGNNs for future node affinity prediction". What precisely do you mean by "incompatible design choice" here? Does this refer back to using memory cells, or are there other unsuitable design choices you are referring to? I also seem to remember, but may be wrong, that not all of the TGNN baselines build on memory cells. If that's the case, can you speculate what could be a reason for their weak performance?
5. I am also wondering whether it has been established that those simple baselines, such as persistent forecasting, generally work well in practice? Clearly, the empirical evaluation shows that this is the case at least in the considered networks. However, are there any other arguments to support this? Do we generally expect node affinities to remain quite stable over time, such that those simple heuristics work well? Are there scenarios where we expect a different behaviour, such that conventional TGNNs would perform better than NAViS?
6. Could you provide an intuitive interpretation of the global vector g? I understand that it maintains previous node affinity vectors, but is that on a per-node basis, or is it a global vector? Would it make sense to think of it similarly to a vector of node centralities?

Minor points
- It seems like the wording in the abstract is somewhat misleading. The abstract states that the authors "introduce a novel loss function for node affinity prediction", which sounds like they designed it from the ground up. However, section 3.3 states that NAViS is trained with the so-called Lambda Loss, introduced by Burges et al. in 2006. Perhaps it would be appropriate to adjust the wording.
- The sentence in l.89 seems to end abruptly: "... they fail to recover the shared latent"
- Typo in l.423 "deign choice"

---

> ### Author Response · Authors · 2025-11-20
> **Part 1 of comment for Reviewer 6w2j**
>
> Dear Reviewer 6w2j, we are excited to read that you find both the theoretical and empirical contributions of our paper valuable, and that you find the manuscript clear.  We are also grateful for your constructive feedback, which we address in our responses below. We have also revised our paper accordingly, with changes marked in blue.
>
>
> 1.**Regarding claim:** Thank you for pointing this out. We agree. In the revised version of our work we removed the second claim and added the following to the first claim: “which have been proven empirically to perform exceptionally well on node affinity tasks”. Thank you.
>
> 2.**Regarding complexity:** We appreciate the constructive feedback. In the “NAVIS for large scale graphs” paragraph in Section 3.2 of the revised manuscript, we now explicitly analyze the memory complexity of NAVIS and discuss how it scales. In Appendix G, we further show empirically that NAVIS scales sub-linearly with the total number of nodes in the graph, and we also include a runtime comparison to other models for both training and inference. Thank you.
>
> 3.**Regarding the question in page 1:** Thank you for the opportunity to elaborate on this important point. Beyond explaining why heuristics currently outperform more sophisticated TGNNs for node affinity prediction, our work also suggests how TGNNs can be pushed further on this task. Specifically, we identify several factors that hinder TGNNs from achieving optimal performance, and NAVIS can be viewed as a concrete baseline that mitigates these issues. By doing so, it can be viewed as a general recipe. We believe these principles can guide the design of improved TGNN architectures for node affinity prediction. Based on your comment and our discussion above, we now elaborate on these directions in the “Limitations and Future Work” paragraph of the revised manuscript. Thank you.
>
> 4.**Regarding expressiveness:** We appreciate the insightful question. Yes, you are correct. For example, NAVIS  cannot (in its current form) express the pointwise nonlinear function (f(x) = tanh(x)), while an RNN cell can. Together with our response to point 3 above, we now clarify this scope in the revised manuscript: in the Limitations and Future Work paragraph, we discuss how one can design an additional nonlinear component for NAVIS, so that it can express such functions without hindering its ability to express the discussed heuristics. Thank you.
>
> 5.**Regarding what NAVIS learns:** This is a very important question which we embrace. To address it, we provide the following experiment to demonstrate that NAVIS can learn the desired heuristics rather than theoretically being able to express them. In Table 2, the Persistent Forecast baseline performs better than Moving Average on tgbn-trade, while on tgbn-genre the opposite holds. This means that tgbn-genre requires more memory to accurately perform node affinity prediction while in tgbn-trade one needs to rely more on the newly arrived data. To examine what NAVIS learns in these two regimes, we measured the average values of $z_h$​ and $z_s$​ from Equation 14, which control the relative influence of the memory component. We expect lower values on tgbn-trade to resemble persistent forecast and larger values on tgbn-genre to resemble moving average. The measured values are $z_h=0.48, z_s​=0.49$ for tgbn-trade and $z_h​=0.90, z_s​=0.84$ for tgbn-genre. This alignment between the learned $z_h, z_s$​ and the heuristic that performs better on each dataset provides empirical evidence that NAVIS learns the intended heuristic behavior in practice. We added this insightful discussion and results to the revised paper in Appendix E. Thank you.
>
> 6.**Regarding Line 423:** By ‘incompatible design choice’ we refer to the combination of architectural and training choices discussed in Sections 3.1 and 3.3, including *(1)* underutilizing the available temporal information, *(2)* the reliance on the standard memory cells, and *(3)* the use of a standard cross-entropy loss that is not aligned with the future node affinity prediction objective. While not all TGNN baselines explicitly use memory cells, they each exhibit at least one of these issues (e.g., limited temporal receptive field or inappropriate training objective), which we conjecture contributes to their weak performance on the task of node affinity prediction. We clarified this notion in the revised manuscript, thank you.

---

> > ### Author Response · Authors · 2025-11-20
> > **Part 2 of comment for Reviewer 6w2j**
> >
> > 7.**Regarding simple heuristics:** Thank you for the important question. In our paper, we worked to address this concern empirically by going beyond the TGB benchmarks, by including additional datasets from the link property prediction literature, where TGNNs are known to be very successful [1,2]. Our findings indicate that even in this setting, simple heuristics still outperform the more complex TGNNs (please see Tables 3-4). Also consider an inductive node affinity prediction setting, where models are tested on nodes that were unseen during training. In this case, the heuristics also offer strong performance: for a persistent forecast model, the first query for a new node is indeed an uninformed guess (because its previous state is reset to zero), but after the first update, the model behaves exactly as in the standard setting. Similarly, for the moving average, the influence of the initial “no-information” state decays over time. Nonetheless, despite our thorough evaluation of heuristics under different settings and tasks, we agree with you that there may exist regimes where conventional TGNNs could outperform them. Following that, we have added a discussion of this potential limitation to the revised manuscript in the “Limitations and future work” paragraph, and view characterizing such regimes as an interesting direction for future work. Thank you.
> >
> > 8.**Regarding the global vector:** The global vector g is aggregated from a buffer that is commonly shared among all nodes. Its goal is to approximate a global trend (e.g, a new song or a new TV series that is globally streamed) before we are queried about a specific node. It can be thought of as a global center when, for example, a mean aggregation is applied to the global buffer. We added this explanation to the revised version of the manuscript.
> >
> > 9.**Regarding wording:** We appreciate your comment. To address it, we changed "introduce a novel loss function for node affinity prediction” to "introduce a dedicated loss function for node affinity prediction” in the abstract. We hope that you find it more appropriate.
> >
> > 10+11. **Regarding typos:** We fixed the typos, thank you.
> >
> >
> > We would like to thank you for the constructive feedback and insightful suggestions and questions, which we find to be beneficial for improving the quality of our paper. If you still have any concerns or questions, please let us know and we will be happy to address them. Otherwise, we would be grateful if you can consider revising your score.
> >
> > **References:**
> >
> > [1] Rossi, Emanuele, et al. "Temporal graph networks for deep learning on dynamic graphs.",2020
> >
> > [2] Yu, Le, et al. "Towards better dynamic graph learning: New architecture and unified library.", 2023

---

> > > ### Comment · Reviewer_6w2j · 2025-11-24
> > >
> > > Thank you for the detailed clarifications, which helped me understand your work better. I find that your revisions, including additional explanations and experiments, have strengthened your paper, and I am raising my score accordingly.

---

> > > > ### Author Response · Authors · 2025-11-26
> > > > **Comment for Reviewer 6w2j**
> > > >
> > > > Dear Reviewer 6w2j,
> > > >
> > > > We thank you for your response and for your feedback, that helped us improving our paper. We are happy to read that you are satisfied with our rebuttal and revisions to the paper, and thank you for your support in our work.
> > > >
> > > > With kindest regards,
> > > >
> > > > Authors.

---

### Official Review · Reviewer_uBoh · 2025-11-01

**Soundness:** 2
**Presentation:** 2
**Contribution:** 2
**Rating:** 2
**Confidence:** 4

**Summary:**

The paper revisits the task of node affinity prediction in temporal graphs, which aims to forecast the future affinity distribution of each node rather than just predicting future links. The authors observe that simple heuristics like Persistent Forecast (PF) and Exponential Moving Average (EMA) often outperform existing Temporal Graph Neural Networks (TGNNs). To address this, they theoretically show that such heuristics can be viewed as linear State Space Models (SSMs) and propose a new TGNN architecture called NAVIS, which integrates a learnable linear SSM with a “virtual global state” to capture global temporal dynamics. The paper also replaces the cross-entropy loss with a ranking-based LambdaLoss to better align with the affinity prediction objective. Extensive experiments on multiple temporal graph benchmarks demonstrate that NAVIS consistently outperforms both heuristic and neural baselines.

**Strengths:**

1. The paper explains why existing TGNNs fail to capture simple temporal dependencies and provides a theoretical basis for improving them.

2. The proposed NAVIS model is conceptually simple yet achieves strong and consistent performance across benchmarks.

**Weaknesses:**

1. There exists a conceptual gap between the theoretical analysis and the modeling assumptions used in practice.
Theorem2 argues that standard RNN/LSTM/GRU units cannot reproduce the persistent forecast (PF, i.e., $h_i = x_i$) because their activation functions restrict the output range. While mathematically valid, this result relies on an assumption that the inputs are unbounded and not normalized. If the inputs are bounded, or if a fixed linear readout is added to counteract activation saturation, the expressive limitation may no longer hold.

2. The design and motivation of the global virtual state $g$ are intuitive but lack precise statistical or identifiability analysis.
The paper only states that $g$ is ``an aggregation of recent affinity vectors in a buffer`` and claims that ``using recent vectors works well in practice``.
However, the following points are not clarified:
(1) What's the specific aggregation operator (mean, exponentially weighted average or  attention？);
(2) There is no statistical or identifiability analysis showing under what conditions this aggregation can truly extract a global trend rather than just averaging noise.
(3) Without (2), it is unclear whether it  models a global latent factor or merely performs temporal smoothing of the previous global affinity distribution.
Since $g$ is central to the claim that heuristics $\approx$ linear SSMs but still require global trend modeling, a deeper ablation across different aggregators and buffer lengths is necessary.

3. Although the authors acknowledge that parameter size grows as $O(N^2)$ and propose truncation by retaining the top-$a$ affinities for each node, there is no quantitative evaluation of the trade-off between $a$, accuracy, and efficiency. In highly dynamic graphs, where new neighbors emerge or distributions shift, truncation based solely on previous affinity magnitudes may omit critical new relations.

4. The main comparisons involve TGNNs and heuristic methods, showing consistent improvements, but no experiments include modern nonlinear, selective SSMs or DyG-Mamba[1,2] under the same protocol. Without such baselines, the claim that ``linear SSM-based TGNNs outperform more complex temporal models`` remains context-dependent and potentially overstated.

5. The authors convert several link-prediction datasets into affinity-prediction datasets by temporal aggregation over daily, yearly, or term-based intervals. This transformation smooths the data and amplifies temporal continuity, which inherently favors linear smoothing models like NAVIS. In scenarios with bursty or non-stationary behaviors, such as cold-start or short-term spikes, the benefit may vanish. Evaluation on finer time resolutions or non-aggregated event streams would better test model robustness to abrupt dynamics.

6. While Table7 reports ablations for three core components, the paper omits finer interpretability analyses: (1) distributions and temporal trajectories of the learned gates $z_h$ and $z_s$; (2) feature-attribution or permutation-importance tests for $g$; and (3) quantitative evidence that $g$ captures regime shifts earlier than node-local dynamics. These analyses would strengthen the claim that the virtual global state encodes shared temporal trends.

7. The paper reports only NDCG@10 and trains on top-20 samples. Metrics such as Recall, MRR should be included for a more complete assessment of affinity prediction quality.

[1] DyGMamba: Efficiently Modeling Long-Term Temporal Dependency on Continuous-Time Dynamic Graphs with State Space Models.

[2] DyG-Mamba: Continuous State Space Modeling on Dynamic Graphs

**Questions:**

Please see the weakness above

---

> ### Author Response · Authors · 2025-11-20
> **Part 1 of comment for Reviewer uBoh**
>
> Dear Reviewer uBoh, we are pleased that you find both the theoretical and empirical contributions of our paper valuable. We now address the concerns that you have raised,and add the incorporated changes in blue to the revised version of the manuscript.
>
> 1.**Regarding Theorem 2:** Thank you for the important discussion. Theorem 2  is indeed proved under the assumption of unbounded data. Nonetheless, we now show that even for normalized input, Theorem 2 holds. Concretely, let us assume the inputs are normalized so that $x_i \in [0,1]^d$. A standard RNN cell is given by $h_i = tanh(W_h h_{i-1} + W_x x_i + b)$. Reproducing the persistent forecast (PF) with the cell output means requiring $h_i = x_i​$ for every possible $x_i \in [0,1]^d$. For this to hold, we must have $tanh(W_h h_{i-1} + W_x x_i + b) = x_i$ for all $x_i \in [0,1]^d$. In particular, PF does not depend on $h_{i-1}$​, so any valid parameter choice must effectively cancel the dependence on $h_{i-1}$​, reducing the mapping to $h_i = \tanh(W x_i + c)$. Let $f(x) = Wx + c$. Then, we need $tanh(f(x))=x$ for all $x \in [0,1]^d$, i.e., $f(x) = \tanh^{-1}(x)$ for all  $x \in [0,1]^d$. However, $f(x)$ is an affine (linear + bias) function, while $tanh^{-1}$ is strictly nonlinear on any interval with nonzero length in [0,1]. Hence, they cannot coincide on $[0,1]^d$. The same applies to $\tilde{h}_i$ in the GRU cell (Eq. 13), and more generally to any standard cell with a non-linear activation in [0,1]: such a cell cannot implement PF on a range of normalized inputs. Following your comment we have also added this proof formally to Appendix H. Thank you.
> Regarding the suggestion of adding a fixed linear readout to counteract activation saturation, we agree that modifying the architecture in this way could, in principle, change the expressiveness of the model. Our theorem is stated for the standard definitions of RNN/LSTM/GRU cells as implemented in widely used libraries (e.g., PyTorch), which do not include such an additional linear readout inside the memory cell itself. Moreover, **prior work we compare to (e.g., TGNv2) also uses these standard cells without such modifications.** This is why we believe *Theorem 2 is practically relevant and meaningful for the graph temporal learning community.*
>
> 2.**Regarding virtual node:** Thank you for the opportunity to elaborate on the global virtual state. We mention in Line 290 that *“In practice, aggregating the buffer with the most recent vector selection…”*, which means that in the main experiments the aggregation operator we use is a selection operator that returns the most recent vector in the buffer. This choice intentionally focuses on the most up-to-date estimate of the global affinity pattern. While it can be noisy because it does not explicitly smooth over past vectors, it provides good results in practice (Table 11). Regarding the statistical/identifiability aspect, our goal with the global virtual state is not to identify the exact global trend in a strict statistical sense, as the end goal of node affinity prediction is to give predictions on the node level. Providing a lightweight approximation for the current global trend that is compatible with the runtime constraints of the heuristics so it can be used as an additional source of information to improve prediction performance in node affinity (L267-L269). *To accommodate your comments and questions*, we have **revised Appendix F**, and we now provide a thorough analysis of different aggregation schemes and buffer sizes, which shows that the recent vector selection achieves a favorable accuracy–efficiency trade-off.
> We agree that accurately identifying global trends in node affinity datasets using advanced aggregation and eviction schemes is an exciting and promising research direction, and we elaborate about it in the “Limitations and future work paragraph” of the revised manuscript.
> Thank you for the insightful questions.

---

> > ### Author Response · Authors · 2025-11-20
> > **Part 2 of comment for Reviewer uBoh**
> >
> > 3.**Regarding model size:** Thank you for the comment. The parameter count of NAVIS grows $O(N)$ and not $O(N^2)$ as we originally stated, since $W_{xh}$,$W_{hh}$, $W_{xs}$, $W_{hs}$, $W_{gs}$​ from Equation 14 are all in $R^{1×d}$, as $z_h$​ and$ z_y$​ are scalars in $[0,1]$. Therefore, we never actually had to use the originally detailed sparsified affinity prediction pipeline. Instead, to reduce the parameter count in large-scale graphs, we set $d$ to be the number of destination nodes. For example, in graphs with users and music genres (tgbn-genre) we only care about the affinities of users to music genres, and not users to other users. **In Appendix G, we show that in practice the number of parameters in NAVIS is often less than the total number of nodes in the graph**, while maintaining accuracy and runtime comparable to or better than the baselines. We have revised the “NAVIS for large-scale graphs” paragraph to reflect this discussion. Thank you.
> >
> > 4.**Regarding SSMs:** Thank you for the opportunity to add an additional comparison to the temporal SSM baseline. In the revised version of our manuscript, we added results on DyGMamba [1] under the same evaluation protocol (please see Table 1 and Table 3). **NAVIS outperforms DyGMamba on all the benchmarks**. We conjecture, based on our empirical analysis, that for node affinity prediction this may be related to the fact that DyGMamba truncates data and does not explicitly capture global temporal dynamics. As demonstrated in the paper, all of these factors can hinder TGNN performance for node affinity prediction. Regarding your concern about the statement that “linear SSM-based TGNNs outperform more complex temporal models,” we carefully re-read the manuscript and could not find this exact claim in the text. We do not intend to make such a broad claim and agree that it would be context-dependent and potentially overstated. To avoid any ambiguity, we further clarify in the revision that our conclusions are restricted to the tasks and benchmarks considered in this work. If you were instead referring to lines 463–464: “A key observation in recent literature is that simple heuristics like moving averages often outperform complex TGNNs on various relevant benchmarks”, we revised it to enhance the clarity, thank you.
> >
> > | Method       | tgbn-trade (Test) | tgbn-genre (Test) | tgbn-reddit (Test) | tgbn-token (Test) |
> > | ------------ | ----------------- | ----------------- | ------------------ | ----------------- |
> > | DyGMamba     | 0.374±0.001       | 0.351±0.001       | 0.314±0.000        | –                 |
> > | NAVIS (ours) | **0.863±0.001**       | **0.520±0.001**       | **0.552±0.001**        | **0.444±0.001**       |
> >
> > (DyGMamaba out of memory in tgbn-token)
> >
> > 5.**Regarding dataset:** We appreciate your comment and agree that temporal aggregation can, in principle, smooth dynamics and potentially favor models with a smoothing bias such as NAVIS. Therefore, we explicitly examined the sensitivity of NAVIS to bursty, short-term phenomena. *First*, we conjecture that short-term spikes and similar noise patterns will not hurt NAVIS significantly. For example, random noise in the ground-truth affinity vector is neglected by Persistent Forecast after consecutive updates to the node state. Similarly, the effect of a spike on Moving Average decays over time, as long as the overall sequence is not dominated by noise. To verify this hypothesis, inspired by your comment, we perturbed the ground-truth affinities of tgbn-genre every 10 and 20 updates by focusing all the affinity on a single random target node. This setting is designed to mimic the short-term spike scenarios you describe. These **results are reported in Appendix E of the revised manuscript**. As reported in Appendix E, these perturbations did not affect NAVIS at all. *Second*, regarding the concern that aggregation may favor NAVIS, we note that our experiments already span a wide range of native temporal granularities. In particular, the dynamic graphs range from 32 steps over 32 years (tgbn-trade) to 21,889,537 steps over 16 years (tgbn-reddit), *according to the official TGB documentation*. We therefore believe that our efforts to perform a fair and comprehensive evaluation of NAVIS and comparing it with other methods highlights its merits. Nonetheless, if there is a specific additional benchmark with substantially different temporal characteristics (e.g., more extreme cold-start or ultra-short-term spikes) that you believe is particularly appropriate for future node affinity prediction, we are happy to add it to our evaluations. Thank you.

---

> > > ### Author Response · Authors · 2025-11-20
> > > **Part 3 of comment for Reviewer uBoh**
> > >
> > > 6.**Regarding ablations:** Thank you for the comment and acknowledging our ablation studies. We welcome your suggestions, and in the revised paper we have included additional interpretability analyses of the virtual global vector. We have also expanded our analysis of the ranking loss function. In addition, in the experiment discussed in Point 5 of *Reviewer 6w2j*, we show that NAVIS can in fact learn when to shift its focus between new and old information.
> > >
> > > 7.**Regarding metrics:** Thank you for bringing up this important point. We chose to present the results in terms of **NDCG@10 because it is the standard metric for TGB benchmarks**. Nonetheless, we welcome your suggestions and in the **revised manuscript we also report MRR and Recall@10 (the fraction of top-10 predicted nodes that appear in top10 ground truth affinity targets)**. These results are provided in Appendix E and are consistent with the conclusions drawn from NDCG@10.
> > >
> > > We would like to thank you once again for you throughout constructive feedback and remarks. We truly believe that the additional analyses and experiments that you suggested, together with strengthening the theoretical contribution based on your observations have significantly improved our paper. If you still have any concerns or questions, please let us know and we will be happy to address them. Otherwise, we will appreciate it if you would consider revising your score.

---

> ### Comment · Reviewer_uBoh · 2025-11-22
>
> Thank you for your response, which has resolved many of my initial concerns, and I have raised my score to 4. However, after a more careful reading and comparison, I now have some additional questions.
>
> 1. In **Theorem 2** of *TGNv2*, the authors also state that *“There exists a formulation of TGNv2 that can represent persistent forecasting and moving average of order k.”* However, in your paper (lines 216–217), you write that *“Tjandra et al., 2024) cannot express the most basic heuristic of PF, thereby hindering memory-based TGNNs’ performance.”* I think you should elaborate further on this point and clarify how your statement differs from the claim made in the original TGNv2 paper.
>
> 2. As I understand it, your method can be viewed as directly applying a linear state-space model (SSM) to address the limitations of TGNN-type models in node-affinity prediction task. You should also compare your approach with other general linear SSMs, since although **DygMamba** incorporates SSMs, it still essentially relies on the CTDG encoder.
>
> 3. To my knowledge, **Persistent Forecast** and **Moving Average** do not require training. Why, then, do you report a *training runtime* for them in **Table 16**? Did I misunderstand something?

---

> > ### Author Response · Authors · 2025-11-23
> > **Comment for Reviewer uBoh**
> >
> > Dear Reviewer uBoh, we are happy to read that many of your initial concerns are resolved, and would like to express our deep gratitude for raising your score. We are also thankful for your response to our rebuttal and additional questions, that we now address below. We have also revised the paper accordingly. We hope that you find our responses satisfactory, and that you will consider revising your score. We also remain available to address any additional questions or comments you may have.
> >
> > 1. **Regarding TGNv2 theory:** Thank you for allowing us to elaborate on this important point. To prove Theorem 2, Tjandra et. al. assume that the memory module incorporated in TGNv2 is a combination of permutation matrices over a specifically designed block matrix and a generator vector (page 10 in the TGNv2 paper). In practice, to implement TGNv2 they utilized a GRU cell as the memory module (page 11 in the TGNv2 paper, and line 169 in the TGNv2-NeurReps/models/mtgn.py file in the official TGNv2 repository). In our paper, we show that using a GRU cell as a memory module does not enable TGNNs to express Persistent Forecast, which helps clarify the gap between theoretical results and practical implementations in prior work. We included this essential remark in the newly revised version of our manuscript in Appendix H, thank you.
> >
> > 2. **Regarding SSM:** Thank you for the important discussion. In our rebuttal, we have included new results in the revised version that compare our NAVIS with DyGMamba, *as you and Reviewer hEbc requested*. In addition, we have originally included results for Persistent Forecast and Moving Average that are also specific types of linear SMM according to Theorem 1. Furthermore, we follow your guidance and we now include an additional comparison in the newly revised manuscript where we compare NAVIS with S4[1], which is a general SSM as in your suggestion. We provide the results here and in Appendix E of the newly revised manuscript. As can be seen, while S4 offers strong performance, our NAVIS outperforms it, further highlighting its effectiveness and contribution. Thank you.
> >
> > | Method       | tgbn-trade (Validation) | tgbn-trade (Test) | tgbn-genre (validation) | tgbn-genre (Test) |
> > | ------------ | ----------------- | ----------------- | ------------------ | ----------------- |
> > | S4    | 0.819±0.005       | 0.796±0.002                 |   0.451±0.001      |      0.461±0.001          |
> > | NAVIS (ours) | **0.872±0.001**  | **0.863±0.001**       |   **0.517±0.001**     | **0.528±0.001**     |
> >
> > 3. **Regarding training times:** Thank you for the comment. Because the standard evaluation protocol for evaluating methods for node affinity prediction follows a chronologically split of the dataset to train-validation-test, all methods, including the Persistent Forecast heuristic and the Moving Average, are required to perform at least one pass over the training data to compute node states right before the validation phase (and later, the test phase) begins. The heuristics, however, only require a single forward pass to compute node states and require no backpropagation steps as suggested in your comment, while other learnable methods require training. We included this important clarification in the newly revised version of our work in Appendix G where we present these results, to enhance clarity. We deeply thank you for this important comment.
> >
> > We would like to express our gratitude for your valuable and thoughtful feedback, which is important for enhancing the quality of our paper. We believe that addressing your additional questions and comments further improved our work. If you have any concerns or questions, we are happy to address them and remain available. We hope that you find our responses in order and that you will consider revising your assessment.
> >
> > **References**
> >
> > [1] Gu, Albert, Karan Goel, and Christopher Ré. "Efficiently modeling long sequences with structured state spaces.",2021

---

> ### Comment · Reviewer_uBoh · 2025-11-26
>
> Thank you for the authors’ response. I think most of the issues have been addressed. I have raised my score to 6. However, I still have a few suggestions for the paper.
>
> 1. Given that the core comparison of the paper is mainly against TGNv2, I think it would make the paper clearer to present TGNv2’s arguments in the **main text, analyze them, and then introduce the paper’s own viewpoint**. This is because the claim that “TGNN cannot express the most basic heuristic of PF, thereby hindering memory-based TGNNs’ performance” is also stated in TGNv2. The main contribution of this paper is, in my view, to refute TGNv2’s theory and then analyze, in practice, why “TGNN cannot express the most basic heuristic” holds.
>
> More specifically, it is too simplistic for the paper to directly state that “TGNN cannot express the most basic heuristic of PF, thereby hindering memory-based TGNNs’ performance” while also implicitly including TGNv2 in this statement. It would be better to discuss TGNv2 separately from other models. For example, you could say: “TGNN cannot express the most basic heuristic of PF, thereby hindering memory-based TGNNs’ performance. Although TGNv2 …, we find that …”
>
>
> 2. In addition, analyzing the differences between the proposed model in this paper and standard SSM models is also an important point. Why do standard SSMs perform worse than the NAVIS proposed in this paper?

---

> > ### Author Response · Authors · 2025-12-02
> > **Comment for Reviewer uBoh**
> >
> > Dear Reviewer uBoh,
> >
> > We would like to express our gratitude for your dedication to the reviewing process. We are glad to know that we managed to address your previous concerns and that you have raised your score accordingly, thank you. Regarding your additional new suggestions for the paper:
> >
> > **Regarding TGNv2:**  Thank you for raising this important point. We welcome your suggestion and incorporate TGNv2’s theoretical results, an appropriate analysis and our paper’s perspective in Subsection 3.1 of the newly revised version of our manuscript. In short, in the TGNv2 paper, Tjandra el. al. shows that TGN is invariant to the identities of the senders and receivers of messages, they also show that this property is undesirable for dynamic node affinity prediction. Then, they solve this by incorporating sender and receiver identities in TGNv2. In our work, we show that identifying senders and receivers is not sufficient alone to express Persistent Forecast. We show that even when the sender and receiver are explicitly identified (as vector entries of the node state), using standard memory cells (RNN, LSTM or GRU) hinders TGNNs from expressing the Persistent Forecast heuristic. Thank you for aiding us to enhance the novelty and contribution of our work.
> >
> > **Regarding SSMs:** Thank you for this suggestion. As previously detailed in point 4 of our original review to you, there are several key factors that explain the difference in performance between our NAVIS and other SSMs, including truncation of relevant information and not capturing global temporal dynamics. We made sure that this notion is clear in the revised version of our manuscript, thank you.
> >
> > Thank you for the support in our work,
> >
> > and best regards,
> >
> > Authors

---

### Official Review · Reviewer_hEbc · 2025-11-01

**Soundness:** 3
**Presentation:** 3
**Contribution:** 3
**Rating:** 6
**Confidence:** 3

**Summary:**

This manuscript addresses the task of future node affinity prediction in continuous time dynamic graphs. Unlike future link prediction, the model output a full ranking / affinity vector over all other nodes. The authors observe that simple heuristics (like persistent forecast and moving average) often outperforms more recent Temporal Graph Neural Networks on this task, and attributes this to four main factors: current TGNN memory cells cannot express even the simplest persistence heuristic; common cross entropy loss is misaligned with the ranking nature of affinity; global temporal dynamics are not captured by local sampling; and batching / truncated buffers cause information loss. To address this, the authors propose NAVIS - TGNN style model built as a linear state space mechanism with both per node and virtual global states. On TGB node affinity and on four converted temporal link datasets, NAVIS outperforms TGNN baselines and the strong heuristics.

**Strengths:**

1. Clearly distinguishes future node affinity prediction (requiring a full ranking) from future link prediction (binary task).
2. Proves that simple heuristics are special cases of linear State-Space Models; and that standard RNN, LSTM, and GRU cells are functionally incapable of representing the persistent forecast heuristics.
3. Extensive experiments on TGB node affinity datasets and four converted link prediction datasets. The results consistently show NAVIS outperforms baselines.

**Weaknesses:**

1. All experiments / comparison is against TGNNs and not against the most recent SSM based temporal models on the same TGBN tasks.
2. Empirical evaluation or ablation study on the performance impact of sparsified affinity prediction pipeline is missing.
3. Evaluation is performed on 4 link prediction datasets that the authors repurposed using a custom pipeline - releasing the exact scripts is important for reproducibility.
4. Analysis of ranking loss is limited to small constructed example. Sensitivity to the margin and to batch size would strengthen the empirical section.

**Questions:**

1. What is the buffer size, eviction policy, aggregation function and normalization? Is g updated per batch or globally across epochs?
2. How are candidate sets constructed? What fraction of true next interactions falls outside the candidate set? Please report recall@K of candidate generation and confirm all methods share the same candidates.

---

> ### Author Response · Authors · 2025-11-20
> **Part 1 of comment for Reviewer hEbc**
>
> Dear Reviewer hEbc, we are happy to know that you find both the theoretical and empirical contributions of our paper valuable, and for your overall positive assessment of our work. We now address the concerns and questions that you have raised, and the incorporated changes can be found colored in blue in the revised version of the manuscript.
>
> 1.**Regarding SSMs:** Thank you for the opportunity to add an additional comparison with a temporal SSM baseline. In the revised version of our manuscript, we have included results on DyGMamba [1] (please see Table 1 and Table 3), a recent SSM-based model for learning on dynamic graphs. *NAVIS outperforms DyGMamba on all the benchmarks*. We conjecture that this is because DyGMamba truncates temporal information and does not fully capture global temporal dynamic. As demonstrated in the paper, these design choices limit performance for node affinity prediction, whereas NAVIS is explicitly designed to address these limitations.
>
> | Method       | tgbn-trade (Test) | tgbn-genre (Test) | tgbn-reddit (Test) | tgbn-token (Test) |
> | ------------ | ----------------- | ----------------- | ------------------ | ----------------- |
> | DyGMamba     | 0.374±0.001       | 0.351±0.001       | 0.314±0.000        | –                 |
> | NAVIS (ours) | **0.863±0.001**       | **0.520±0.001**       | **0.552±0.001**        | **0.444±0.001**       |
>
> (DyGMamaba out of memory in tgbn-token)
>
>
> 2. **Regarding ablation:** Thanks for your important comment. The parameter count of NAVIS actually grows as $O(N)$, not $O(N^2)$ as we previously stated, because $W_{xh}$,$W_{hh}$,$W_{xs}$,$W_{hs}$,$W_{gs}$​ from Equation 14 are all in $R^{1×d}$, since $z_h$​ and $z_y$ are scalars (in [0,1]). Consequently, the sparsified affinity prediction pipeline we originally detailed is not used in any of our experiments. Instead, to keep the parameter count manageable on large-scale graphs, we set $d$ to be the number of all possible destination nodes. In Appendix G we show that, in practice, the number of parameters in NAVIS is often smaller than the number of total nodes in the graph, demonstrating its scalability without requiring the additional sparsified pipeline. We have revised the “NAVIS for large-scale graphs” paragraph accordingly in the new version of our work. Thank you.
>
> 3. **Regarding scripts:** We absolutely agree that releasing the exact scripts for repurposing link prediction datasets to node affinity prediction datasets is important for reproducibility. To this end, we have already provided the exact scripts used in our experiments in the anonymous GitHub repository, under the link2node directory, including a step-by-step guide on how to integrate them into the TGB framework. The link to the anonymous repository appears in the abstract of the paper. Upon acceptance, we will release the code in a public repository as well.
>
> 4. **Regarding ranking loss:** We embrace your suggestion. Following your guidance, we have expanded our empirical analysis beyond the existing constructed example and now include a detailed sensitivity study of the ranking loss with respect to both batch size and the margin (delta) in **Appendix F of the revised manuscript**. Thank you.
>
> 5. **Regarding settings:** Thank you for the question. Because we chose an aggregation scheme that always uses the most recent vector in the buffer, a buffer of size 1 is sufficient. Concretely, when a new node state is computed, it overwrites the previous entry in the buffer (capacity-1 buffer with a trivial eviction policy: always evict the older vector). Thus, the aggregation function over the buffer reduces to simply taking this most recent vector. Because affinity scores are already normalized by TGB no further normalization is required. Following your comment, we additionally included an analysis of different aggregation schemes and buffer sizes in Appendix F of the revised manuscript.

---

> ### Author Response · Authors · 2025-11-20
> **Part 2 of comment for Reviewer hEbc**
>
> 6.**Regarding candidate set:** The candidate set is a property of each dataset and consists of all nodes that ever appear as targets (i.e., nodes that have at least one source node with an affinity/interaction to them). For example, in tgbn-genre the graph contains both users and music genres, but only music genres ever appear as targets, so only music genres are included in the candidate set. By construction, every ground-truth next interaction has its target node in this set, so the fraction of true next interactions that fall outside the candidate set is 0% (i.e., the recall of the candidates  is 100%). All methods are evaluated on exactly the same candidate sets and never assign affinity scores to nodes that are not in the candidate set. This is enforced by the TGB framework. We additionally include results in terms of recall@10 (how many of the top 10 ground-truth affinity targets are ranked in the top-10 predicted nodes) in Appendix E of the revised manuscript.
>
> We genuinely believe that the additional discussions, analyses, and experiments that you suggested helped us improve our paper. If you still have any concerns or questions, please let us know. Otherwise, we will appreciate it, if you can consider raising your score. Thank you.
>
> **References:**
>
> [1] DyGMamba: Efficiently Modeling Long-Term Temporal Dependency on Continuous-Time Dynamic Graphs with State Space Models

---

### Author Response · Authors · 2025-12-02
**Comment for Area Chairs**

Dear Area Chair,

We would like to express our sincere gratitude for your commitment to the reviewing process. We genuinely believe that the reviews we received were highly constructive and beneficial, and that revising our paper according to them improved the quality of our work. All our changes to the paper appear in our current revision on OpenReview, and are marked in blue. In our comment below, we provide a summary of our rebuttal and the author-reviewer discussion period, for your convenience. Thank you.
**Reviewer uBoh** commented that we had resolved many of their original concerns and thus initially raised the score to 4.  In the follow-up discussion, Reviewer uBoh provided additional follow-up questions. We addressed these questions, and **Reviewer uBoh** commented that the relevant issues had been adequately addressed, and **raised the score to 6**. We incorporated the final suggestions of Reviewer uBoh in our revised manuscript. Reviewer uBoh could not comment further at this stage due to the ICLR blocking of the reviewers.
**Reviewer 6w2j** commented that the revisions we made based on all the reviews had strengthened our work, and **raised the score to 8**.
**Reviewers hEbc and QArm** did not have the chance to participate in the early termination of the discussion period, and their original score remained positive.
At the end of this author–reviewer discussion, all reviewers agreed that our work should be accepted, with final scores **6, 6, 6, 8**. We are grateful to each reviewer for helping us improve our work significantly through their thoughtful and detailed feedback.

---
Besides the summary on the scores received from the reviewers, below, we list the major revisions we made to our work based on the reviews and comments received. In the revised version of the paper, we marked these revisions in blue font:


* We added a Theoretical Remarks section in Appendix H, where we formally show that even when the input is normalized, standard memory cells cannot express the Persistent Forecast heuristic. In this section, we also elaborate on the proof of Theorem 3. We also distinguish our Theorem 2 from Theorems 1 and 2 presented in the TGNv2 paper. (Reviewer uBoh and Reviewer QArm)


* We added additional results and explanations for SSMs (DyGMamba). The results can be found in Table 1, Table 3, and Table 11. (Reviewer uBoh and Reviewer hEbc)

* We added additional analysis of NAVIS hyperparameters, including the loss function parameters and the global virtual vector parameters. (Reviewer hEbc and Reviewer uBoh)


* We added a new time and memory comparison and discussion in the “NAVIS for large-scale graphs” paragraph in Subsection 3.2 and in Appendix G, and clarified that the heuristic requires training runtime to compute node states for later evaluation phases (validation and test). (Reviewer hEbc, Reviewer uBoh and Reviewer QArm)


* We performed a noisy-dataset experiment to examine the effect of signal peaks on NAVIS. (Reviewer uBoh)


* We reported performance in terms of MRR and Recall@10, in addition to the original NDCG@10 metric. (Reviewer hEbc, Reviewer uBoh and Reviewer QArm)


* We performed an experiment to examine what NAVIS actually learns. (Reviewer 6w2j)


* We added a “Limitations and future work” paragraph. (Reviewer 6w2j and Reviewer QArm)


* We added a discussion regarding temporal link prediction on weighted dynamic graphs in the related work section. (Reviewer QArm)


* We relaxed some of the claims, as requested by the reviewers. (Reviewer uBoh and Reviewer 6w2j)


* We fixed typos found by the reviewers, as well as other minor issues.  (Reviewer 6w2j)

---

Finally, we would once again like to thank you, our reviewers, for your in-depth reviews. We are also thankful for our Area Chairs both the original and newly assigned chair, and we feel that revising our work according to the reviewers’ comments has improved its quality. As commented by Reviewer uBoh and Reviewer 6w2j, we strongly believe that our revised version has properly addressed all the concerns and questions raised by the reviewers.


Thank you, and best regards,

Authors.

---

### Meta-Review · Area_Chair_6NLj · 2025-12-10

**Summary:**

This paper studies node affinity prediction on temporal graphs, explains why simple forecasting heuristics can outperform common temporal graph neural networks, and proposes NAVIS, a linear state space based model with a virtual global state and a ranking oriented loss that achieves strong gains on several benchmarks. Reviewers initially raised concerns about the strength and scope of the theoretical claims, the role and interpretation of the global virtual state, missing comparisons to modern state space models, incomplete complexity and hyperparameter analyses, and limited discussion of related work on weighted dynamic graphs. Through the rebuttal and revision, the authors added stronger proofs under realistic assumptions, new experiments with DyGMamba and S4, sensitivity studies and efficiency comparisons, more interpretation of what NAVIS learns in practice, and a clearer positioning relative to TGNv2 and temporal link prediction on weighted graphs, which led reviewers to raise their scores and converge on acceptance.

**Reviewer Concerns:**

The authors provided strengthened theoretical arguments, clarified the assumptions behind Theorem 2, and added comparisons to DyGMamba and S4, which resolved requests for broader baselines. They also expanded the analysis of the global virtual state, added sensitivity studies and efficiency measurements, and clarified implementation details, which addressed questions about scalability, loss design, and interpretability. What remains only partially addressed are higher-level issues about the broader scope of the theoretical claims and the extent to which NAVIS differs from or generalizes standard state space models; these points are acknowledged in the revised discussion but are not explored in depth.

**Reviewer Scores:**

Reviewer uBoh began at 2 and later increased the score to 4 and then to 6 after the additional theoretical clarifications, new SSM comparisons, and expanded analyses. Given these explicit statements, it is reasonable to assume the reviewer would have kept the final higher score. Reviewer hEbc started at 6 and did not participate further, but the revisions added all requested experiments and clarifications, so the score would likely have remained unchanged. Reviewer 6w2j began at 6 and explicitly raised the score to 8 after the authors addressed questions about expressiveness, learned behavior, and complexity, so this score would have held. Reviewer QArm started at 6 and did not rejoin the discussion; the authors addressed all listed concerns in detail, so the score would likely have stayed at 6.

---

### Decision · Program_Chairs · 2026-01-26

Accept (Poster)